# Mapping the Link between Climate Change and Mangrove Forest: A Global Overview of the Literature

Thirukanthan Chandra Segaran [1,*,†], Mohamad Nor Azra [1,2,*,†], Fathurrahman Lananan [3], Juris Burlakovs [4], Zane Vincevica-Gaile [5], Vita Rudovica [6], Inga Grinfelde [7], Nur Hannah Abd Rahim [8] and Behara Satyanarayana [8,9,10]

1   Institute of Marine Biotechnology (IMB), Universiti Malaysia Terengganu (UMT), Kuala Nerus 21030, Terengganu, Malaysia
2   Research Center for Marine and Land Bioindustry, Earth Sciences and Maritime Organization, National Research and Innovation Agency (BRIN), Pemenang 83352, Indonesia
3   East Coast Environmental Research Institute, Universiti Sultan Zainal Abidin, Gong Badak Campus, Kuala Nerus 21300, Terengganu, Malaysia
4   Mineral and Energy Economy Research, Institute of the Polish Academy of Sciences, 31-261 Krakow, Poland
5   Department of Environmental Science, University of Latvia, LV-1004 Riga, Latvia
6   Department of Analytical Chemistry, University of Latvia, LV-1004 Riga, Latvia
7   Laboratory of Forest and Water Resources, Latvia University of Life Sciences and Technologies, LV-3001 Jelgava, Latvia
8   Mangrove Research Unit (MARU), Institute of Oceanography and Environment, Universiti Malaysia Terengganu, Kuala Nerus 21030, Terengganu, Malaysia
9   Systems Ecology and Resource Management Research Unit, Département de Biologie des Organismes, Université Libre de Bruxelles, B-1050 Brussels, Belgium
10  Mangrove Specialist Group (MSG), Species Survival Commission (SSC), International Union for the Conservation of Nature (IUCN), c/o Zoological Society of London, London NW1 4RY, UK
*   Correspondence: thiru@umt.edu.my (T.C.S.); azramn@umt.edu.my (M.N.A.); Tel.: +60-9668-3675 (T.C.S.); +60-9668-3785 (M.N.A.)
†   These authors contributed equally to this work.

**Abstract:** Mangroves play a crucial role in maintaining the stability of coastal regions, particularly in the face of climate change. To gain insight into associations between climate change and mangroves, we conducted bibliometric research on the global indexed database of the Web of Knowledge, Core Collection. A total of 4458 literature were analyzed based on bibliometric information and article metadata through a scientometric analysis of citation analysis as well as a cluster analysis. Results suggest that coastal countries such as the USA, Australia, China, India, and Brazil are showing the recent influential mangrove-related keywords such as blue carbon and carbon stock. Interestingly, the "carbon stock", "Saudi Arabia", "range expansion" and "nature-based flood risk mitigation" is among the top cluster networks in the field of climate change and mangrove forest. The present research is expected to attract potential leaders in research, government, civil society, and business to advance progress towards mangrove sustainability in the changing climate meaningfully.

**Keywords:** carbon stock; diversity distribution; ecosystem; heavy metals; forest; land–climate interactions; palynology; *Rhizophora*; sea level rise; tidal current

## 1. Introduction

Mangroves are referred to as plants or trees as well as being defined in the context of the mixture of various plant types or called a "tidal forest" or a "mangrove forest" [1]. Mangroves are valuable economic and ecological resources, being a breeding ground for fish, shellfish, and birds, as well as being a renewable source of wood and offering protection against climate change, especially coastal erosion [2–8]. Most of the mangroves are also physiologically adapted to various anthropogenic problems, such as anoxia, high salinity, and frequent tidal flooding. The scientific community has conducted extensive research

on mangroves covering a variety of topics. This includes the origin, biology, and growth history of mangroves [9], biogeographic distribution and biodiversity of mangroves [10,11], physical characteristics and hydrodynamics [12,13], rehabilitation, restoration and conservation strategies of mangrove ecosystem [14,15], mangrove mapping using remote sensing [16–18], chemical characteristics and carbon sequestration [19–21], a mangrove ecosystem and its associated biodiversity [22,23], secondary metabolites and valuable bioactive compounds derived from a mangrove forest [24,25], deforestation and exploitation of a mangrove forest [26], economic analysis and management strategies [27], the impact of climate change, and nutrient and soil pollution on the mangrove ecosystem [28,29]. There have also been several highly cited review articles on mangroves, such as the review on mangrove ecosystems and rehabilitation [14], mangrove carbon dynamics [30], socio-economics, ethnobiology and management of mangroves [31], carbon cycling and storage [32], and the impact of rising sea levels on mangroves [33].

Currently, bibliometric information with article metadata has been widely used as a tool for research-on-research performance and perspective [34–38]. Scientometric analysis is one of the methods to monitor emerging trends and research progress in selected areas. CiteSpace is one of the most widely used softwares associated with scientometric techniques. Few metrics in CiteSpace can be considered important for any document or citation analysis in the field. Centrality and burstiness detection is a metric to identify and measure the importance of a document in a field within the networks [39]. Any detected publication with high centrality and burst detection can be considered the most influential literature in the field. The research topics in the discussion section will be selected based on these criteria. Prominent groupings generated by the CiteSpace through the extraction of noun phrases from the titles, keyword lists, or abstracts in all selected articles are also one of the simple and useful indicators to identify the cluster available in the area.

There have also been several publications using bibliometrics and scientometric analysis on mangroves and their associated ecosystem, such as the publication trends in mangrove forests [40], tropical mangrove forest land-use [41], analysis on mangrove related publications [42], mangroves as blue carbon sinks [43], mangrove-derived bioactive compounds to combat neurodegenerative diseases [44], and many more. This demonstrates the breadth of issues surrounding mangrove research, as well as the need to employ bibliometrics, especially scientometric analysis, to obtain a holistic perspective of the field as a whole. Thus, this review aims to address the key question: "How have mangrove forests been affected by climate change and its interactions"? Specific objectives were to assess the mangrove-related literature towards changing climate in terms of (i) annual number of articles published, (ii) countries/regions involved in the field, (iii) research topics, (iv) co-cited networks (i.e., frequency of two different documents are cited together in other documents) (v) cluster networks, (vi) research topic (i.e., keywords) burstiness, (vii) dual map overlay, and (viii) the future trends of the knowledge domain (i.e., mangrove and climate change).

## 2. Materials and Methods

### 2.1. Bibliometric Analysis

Clarivate Analytics' Web of Science (WOS) platform was used to generate the metadata for the present study, and Core Collection (WOSCC) database was the only main dataset used for the purpose. There are two different keywords used to perform the search activities: "mangrove" and "climate change". For the mangrove, the appropriate synonym is ("mangrove*") OR ("mangrove* forest") OR ("tidal forest"), which is based on the previous studies by Ho and Mukul [40], Mohd Razali et al. [41], and Saravanan and Dominic [42]. For the climate change related elements, the keywords were based on the modifications by Azra et al. [45], which are ("climat*") OR ("climat* chang*") OR ("global warm*") OR ("seasonal* variat*") OR ("extrem* event*") OR ("environment* variab*") OR ("anthropogenic effect*") OR ("greenhouse effect*") OR ("sea level ris*") OR (erosio*) OR ("agricult* runoff") OR ("weather* variab*") OR ("weather* extrem*") OR ("extreme*

climat*") OR ("environment* impact*") OR ("environment* chang*") OR ("anthropogenic stres*") OR ("temperature ris*") OR ("temperature effect*") OR ("warm* ocean") OR ("sea surface* temperat*") OR (heatwav*) OR (acidific*) OR (hurrican*) OR ("el nino") OR ("el-nino") OR ("la nina") OR (la-nina) OR (drought*) OR (flood*) OR ("high precipit*") OR ("heavy rainfall*") OR ("CO2 concentrat*") OR ("melt* of the glacier*") OR ("melt* ice*") OR ("therm* stress*") OR ("drought") OR ("hypoxia"). Full bibliometrics records and cited references from WOSCC were downloaded in the Plain Text File (.txt) format, in which each download was limited to only 500 data entries.

### 2.2. Data Visualization

Figure 1 indicates the framework for the scientometric review based on the general bibliometric methods. The scientometric software of CiteSpace generated the visualized graph. CiteSpace version 6.1.R6 (64-bit) for Advance users was used on 26 December 2022. In Citespace, a few features were usually revealed through the visualization of charts created automatically by the algorithm of the software. These features allow for a clear and effective presentation of the downloaded data from WOSCC. In CiteSpace, time was set automatically based on the generated database (1977–2021), with a 1-year time slice as well as Top N (50 levels) as a selection criterion of most cited or occurred items from each slice (i.e., year).

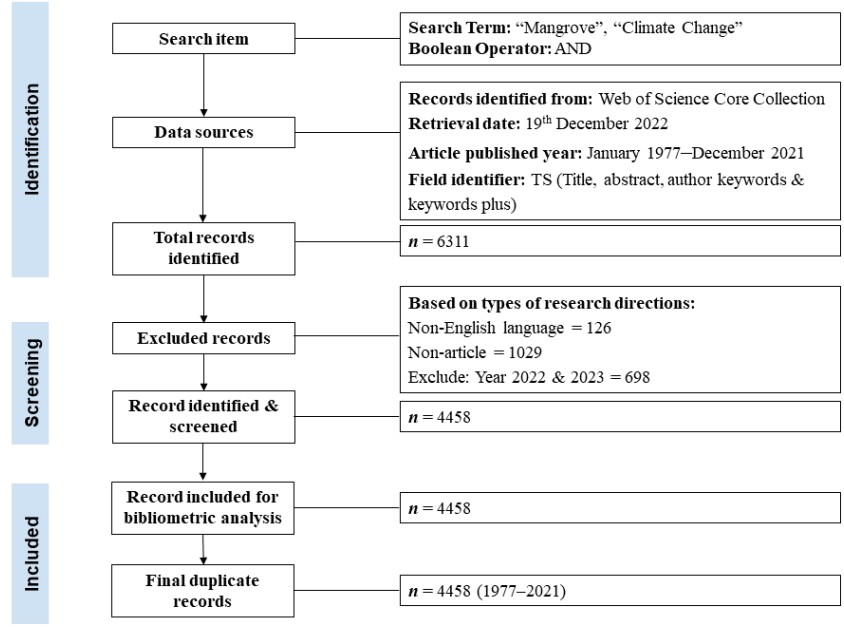

**Figure 1.** The framework of the present study of climate change and mangrove forests.

## 3. Results

### 3.1. Annual Publication Trends and Productive Journals

The search identified 4458 papers related to the impact of climate change on mangroves between 1977 and 2021 (Figure 2). The earliest article relating the climatic impact on mangroves was the study titled "Reconstructing Triassic vegetation of eastern Australasia", which was published in 1977 by Retallack [46], and which stood for the pioneering approach in dating and mapping fossil plant associations, which included Pachydermophylletum (a mangrove scrub) and Linguifolietum (coastal swamp woodland) from the Pacific margin of Gondwanaland during Middle Triassic time. There has been a strong upsurge in publications relating to topics of interest during the past decade, suggesting that academia is paying increasing attention to this subject. We found that 77.2% of the datasets, or 3517 papers, are from the past decade (2011–2021), with 2283 of those papers alone appearing in the last five years (2017–2021). All the articles recorded on mangrove forest-related studies in this review were distributed in 911 journals. *Estuarine, Coastal and*

*Shelf Science* ranked first in the number of publications (221), followed by *The Journal of Coastal Research* (113), *Science of the Total Environment* (85), *Ocean & Coastal Management* (79) and *Estuaries and Coasts* (78), (Table 1).

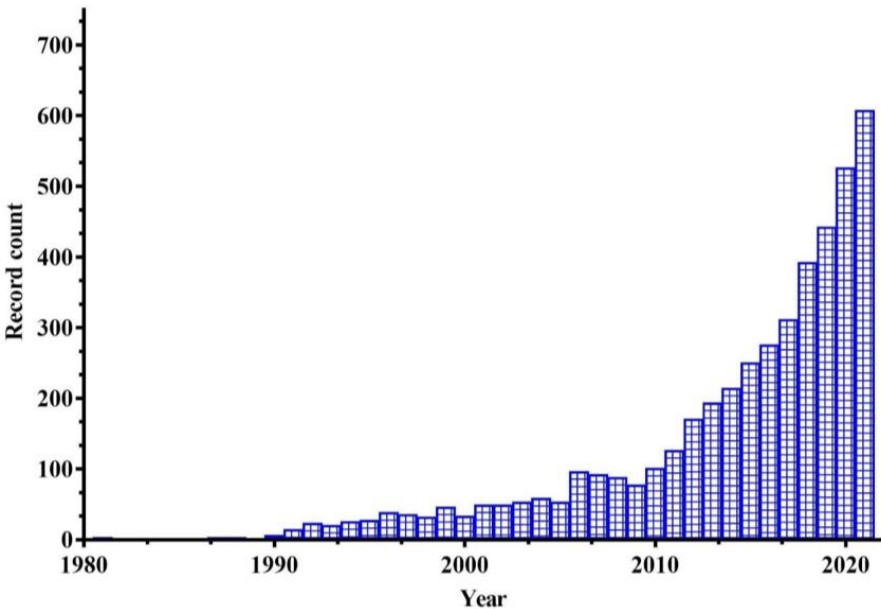

**Figure 2.** The number of original research articles on the impact of climate change on mangrove forest-related studies published between 1977 and 2021.

**Table 1.** Top 10-most productive journals publishing papers on the impact of climate change on mangroves.

| Journals | Quartile (2021) | Impact Factor (2021) | Record Count |
|---|---|---|---|
| *Estuarine, Coastal and Shelf Science* | Q1 | 3.10 | 221 |
| *Journal of Coastal Research* | Q3 | 0.67 | 113 |
| *Science of the Total Environment* | Q1 | 10.15 | 85 |
| *Ocean & Coastal Management* | Q1 | 4.33 | 79 |
| *Estuaries and Coasts* | Q1 | 2.78 | 78 |
| *PLoS ONE* | Q1 | 3.58 | 77 |
| *Hydrobiologia* | Q1 | 2.60 | 71 |
| *Marine Pollution Bulletin* | Q1 | 6.49 | 65 |
| *Remote Sensing of Environment* | Q1 | 13.63 | 64 |
| *Wetlands* | Q2 | 2.00 | 61 |

*3.2. International Cooperation, Collaborative Network and Funding Bodies*

The countries involved in mangrove research, particularly looking into the impact of climate change, shows a striking disparity in the distribution of articles between continents. A total of 139 countries were detected (Figure 3). In terms of geographic distribution, the United States (1328 papers), Australia (734 papers), and China (477 papers) together accounted for more than 50% of the total number of publications (*n* = 4458). India, Brazil, Germany, Mexico, and England are next in order.

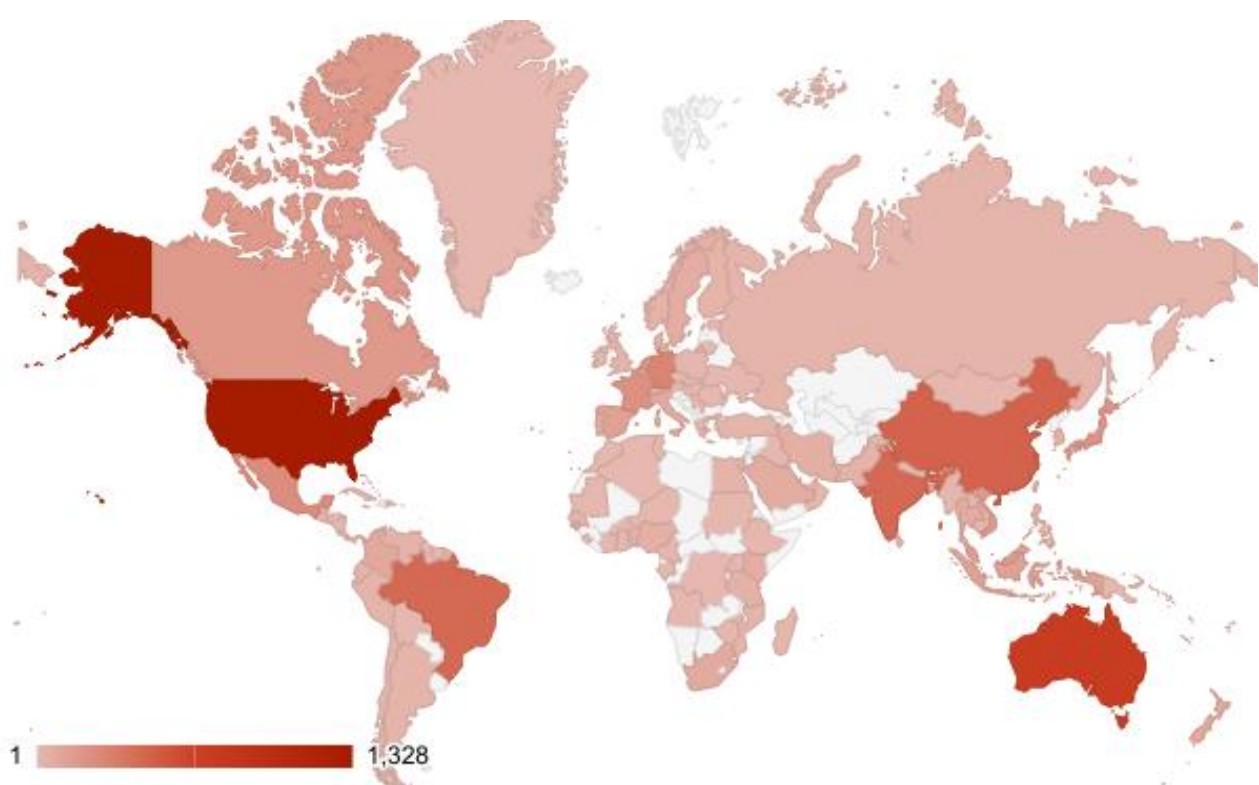

**Figure 3.** The total number of research publications on the impact of climate change on mangroves per country on mangrove forest-related studies. The darker orange reflects the greater number of manuscripts, while lighter shades reflect a moderate number to fewer publications.

Research scholars' cooperation in generating new scientific knowledge is called scientific collaboration [47]. Mangrove-related studies have specific geographical attention. Most researchers adopt the principle of proximity in the sample selected for their studies, so the geographical distribution among countries can reflect the depth and breadth of research on relevant aspects in different regions. There are 137 nodes and 852 linked lines in the country collaboration mapping from CiteSpace (Figure 4). The largest extent of mangroves is found in Asia (40.8%), followed by the Americas (30.4%), Africa (18.3%), and Oceania (10.4%) [48]. While the world's remaining mangrove forests are spread across 118 countries, approximately 75% of mangroves are concentrated in just 15 countries which include Indonesia, Brazil, Australia, Mexico, Nigeria, Malaysia, Myanmar, Bangladesh, Cuba, India, Papua New Guinea, Guinea Bissau, Mozambique, Madagascar, and Philippines [49]. This is reflected in the country cooperation network analysis, where there are definitely large quantities of publications coming out from these 15 countries based on the size of the nodes. It was seen that some of the major contributors in the field of mangrove-related research were countries from the European Union, Japan, and England, despite having a meagre to no mangrove cover. This could be associated with the availability of funding from these countries to conduct research together with their partnering institutions worldwide (Table 2).

The National Science Foundation led the funding agencies for topics related to mangroves vis-à-vis climate change with the highest number of research publications—with 293—closely followed by the Natural Science Foundation of China (NSFC), supporting 283 publications (Table 2). The names and numbers of the top 10 funding agencies were from the United States of America, China, Australia, Brazil, Mexico, the United Kingdom, Japan, and the European Union.

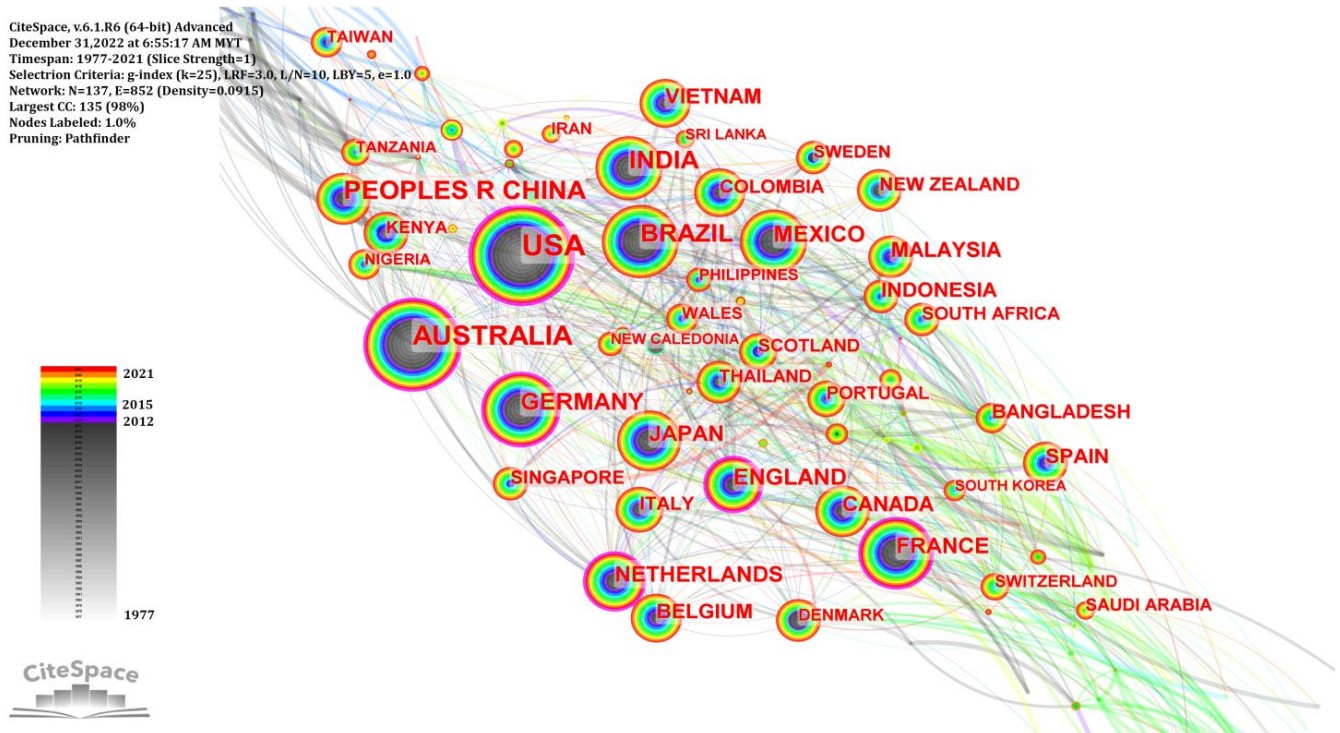

**Figure 4.** An interactive illustration of the cooperation network among countries that have published research on the impact of climate change on mangroves from 1977–2021. The lines connecting the nodes represent the strength of connections between research fields, with thicker lines indicating a higher intensity of connections. The size of the nodes signifies the frequency of co-occurrence in the research fields. The nodes are color-coded to represent the year of publication, e.g., red representing 2021, yellow representing 2019, blue representing 2015, and purple representing 2012.

**Table 2.** Top 10 major grant providers related to climate impact on mangrove research.

| Funding Agencies | Total Number of Grants |
|---|---|
| National Science Foundation (NSF) | *293* |
| National Natural Science Foundation of China (NSFC) | *283* |
| National Council for Scientific and Technological Development (CNPq) | *199* |
| Australian Research Council | *190* |
| Coordination for the Improvement of Higher Education Personnel (CAPES) | 129 |
| National Council of Science and Technology (CONACYT) | 91 |
| São Paulo Research Foundation (FAPESP) | 84 |
| UK Research and Innovation (UKRI) | 84 |
| Ministry of Education, Culture, Sports, Science & Technology | 74 |

### 3.3. Important Research Disciplines

CiteSpace's "Category" node type was used to generate a visualization map showing the disciplinary categories represented by 4458 papers analyzed in this study. The correlation burst detection analysis can detect the activity of research disciplines. After being simplified by pathfinder network scaling, an 86-node network of research discipline co-occurrence from 1977 to 2021 was obtained (Figure 5). The studies on the climatic impact on mangroves have been multifaceted through different disciplines, as demonstrated by the network map (Figure 5). The five research disciplines include Environmental Sciences, Marine Freshwater Biology, Ecology, Geosciences Multidisciplinary, and Oceanography. En-

vironmental Sciences, the largest contributor with 31.5% of the total publications, provides a comprehensive understanding of the impacts of climate change on mangrove forests in relation to broader environmental issues. Marine Freshwater Biology, with 978 publications, focuses on the biological implications of changes in sea levels and salinity levels on mangrove forests and associated species. Ecology, with 812 publications, investigates the role of mangrove forests in coastal ecosystems and the consequences of their decline as a result of climate change. Geosciences Multidisciplinary, with 735 publications, examines mangrove forests' physical and geological aspects, including their response to sea-level rise and increased extreme weather activities. Finally, with 630 publications, Oceanography explores the oceanographic processes affecting mangrove forests, such as changes in water temperature, circulation patterns, and wave patterns. These five research disciplines provide a comprehensive understanding of the impact of climate change on mangrove forests, integrating the environmental, biological, physical, and oceanographic perspectives.

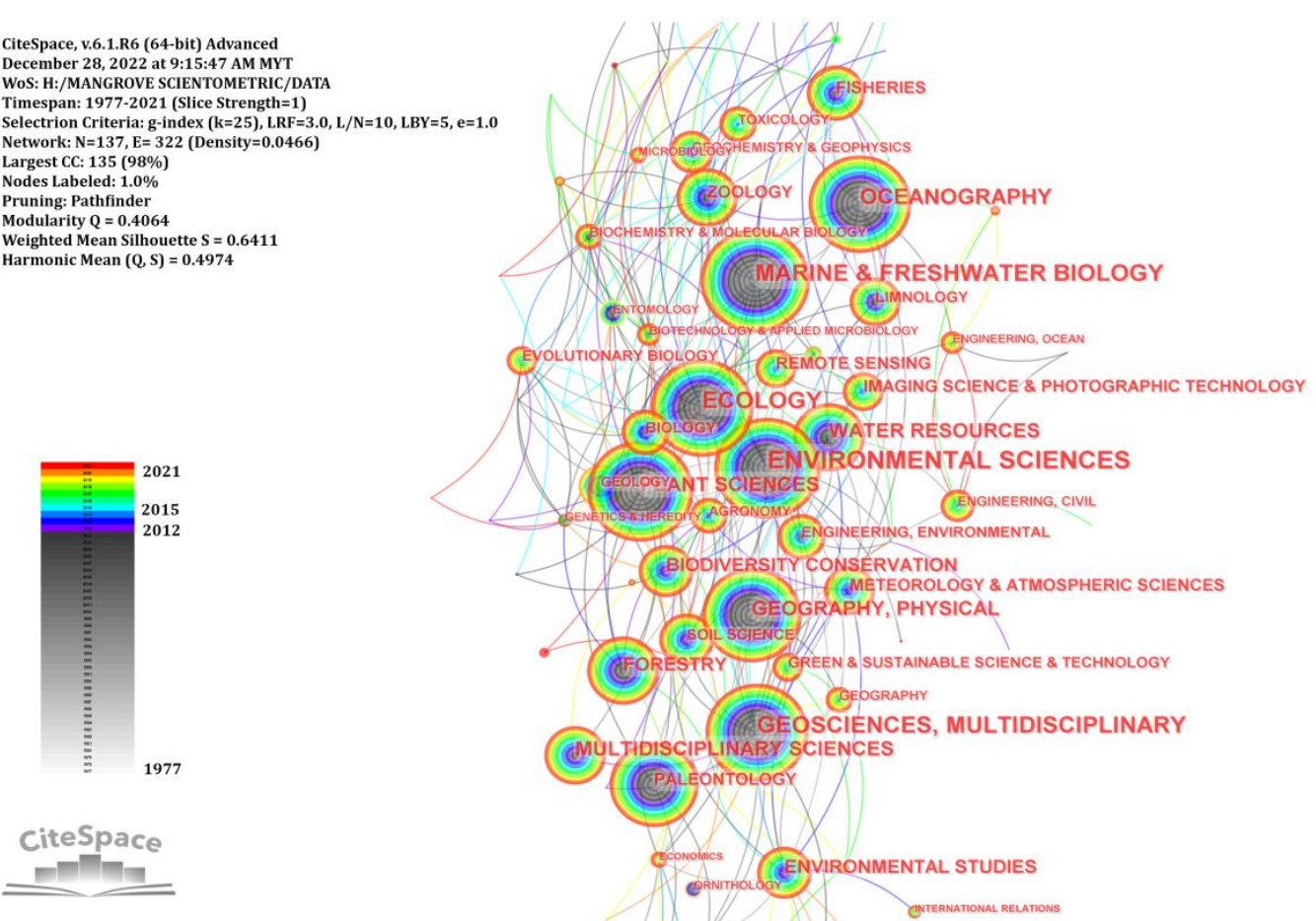

**Figure 5.** Network links between research disciplines. The lines connecting the nodes represent the strength of connections between research fields, with thicker lines indicating a higher intensity of connections. The size of the nodes signifies the frequency of co-occurrence in the subject category. The nodes are color-coded to represent the year of publication, e.g., red representing 2021, yellow representing 2019, blue representing 2015, and purple representing 2012.

### 3.4. Research Cluster Analysis

Cluster analysis is a popular method of statistical data analysis and knowledge discovery because of its ability to uncover latent semantic themes in textual data [50,51]. Cluster analysis can divide a large body of research data into various units based on the relative degree of term correlation, making it easier to identify the research themes, trends, and connections within a given field of study [51,52]. A cluster's homogeneity can be quantified

using an index called the mean silhouette, with values ranging from −1 to 1. The average silhouette value for each cluster was determined using CiteSpace. The higher the value, the more similar the cluster's members are to one another [53]. The network showed 42 clusters in the context of the scientometric analysis mapping the link between climate change and mangroves (Table 3) and (Figure 6). In conducting this analysis, we employed the two primary algorithms available in Citespace, namely the Log-Likelihood (LLR) and Latent Semantic Index (LSI) algorithms. The LLR algorithm assesses the similarity between the text content and topics, while the LSI algorithm categorizes technical terminologies. The literature referenced by the citing literature from 1977 to 2021 served as the data source for the scientometric study carried out using the co-citation network of cited references, as shown in Figure 7.

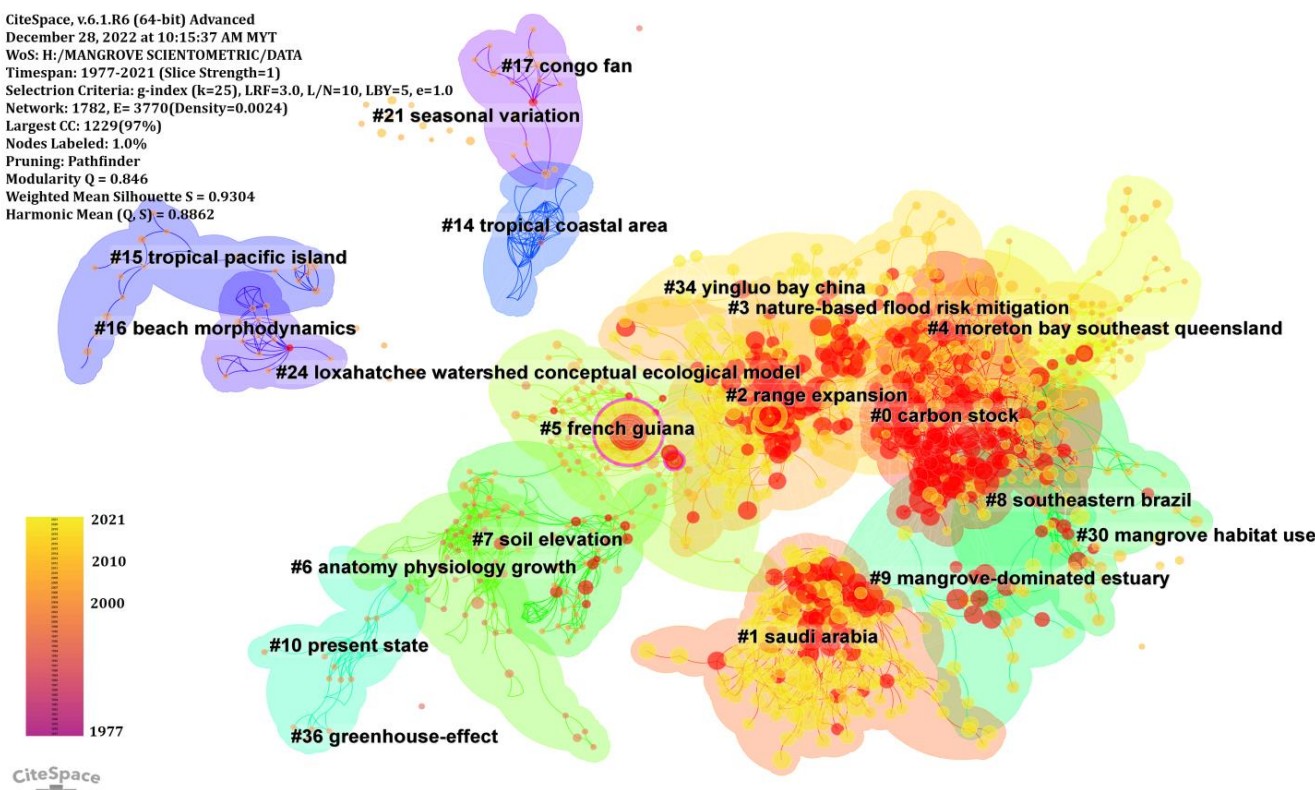

**Figure 6.** The reference co-citation network of publications on mangroves and climate change from 1977 to 2021 was analyzed. The size of the nodes in the network reflects the frequency of citation, while the colors of the nodes, ranging from magenta (1977) to yellow (2021), indicate the progression of research over time. The colored connections represent co-citation relationships. The network was further divided into 42 clusters through network clustering analysis.

The largest cluster (#0) has 207 members and a silhouette value of 0.908. It is labelled as carbon stock by LLR and mangrove forest by LSI. The second largest cluster (#1) has 194 members and a silhouette value of 0.904. It is labelled as Saudi Arabia by LLR and mangrove forest by LSI. The third largest cluster (#2) has 159 members and a silhouette value of 0.874. It is labelled as range expansion by LLR and coastal wetland by LSI. The major citing article of all these three clusters is by Osland et al. [54], where they described the impacts on conservation strategies in the mangrove forests along the coast of the Gulf of Mexico. In their conclusion, they noted that changes in precipitation rate, hydrological, and estuarine regimes would be the most significant factors impacting mangroves in this region around the turn of the century. In addition to climatic variations, accelerated sea-level rise, intensified tropical cyclones, elevated carbon dioxide, land use change, eutrophication, and invasive non-native species are anticipated to have a disproportionately large impact on

mangrove forests by altering inundation and salinity regimes, which have a substantial impact on the structure and function of mangrove ecosystems.

**Table 3.** Top-ranked clusters and labels produced by LSI and LLR on mangrove forest-related studies.

| C * | Si¹ | Si² | Yr | Label (LSI) | Label (LLR) |
|---|---|---|---|---|---|
| 0 | 207 | 0.908 | 2010 | Mangrove forest | Carbon stock |
| 1 | 194 | 0.904 | 2016 | Mangrove forest | Saudi Arabia |
| 2 | 159 | 0.874 | 2015 | Coastal wetland | Expansion |
| 3 | 124 | 0.908 | 2014 | Sea-level rise | Sea-level rise |
| 4 | 87 | 0.951 | 2006 | Sea-level rise | Moreton Bay Southeast Queensland |
| 5 | 84 | 0.925 | 2003 | Northern Brazil | French Guiana |
| 6 | 68 | 0.993 | 1993 | *Rhizophora mangle* | Anatomy physiology growth |
| 7 | 64 | 0.984 | 2002 | Soil elevation | Soil elevation |
| 8 | 41 | 0.971 | 2009 | Holocene mangrove dynamics | South-eastern Brazil |
| 9 | 37 | 0.971 | 2016 | Mangrove dominated estuary | Mangrove-dominated estuary |

\* C- represents Cluster; Si¹ represents size; Si² represents silhouette; Yr represents the mean of the cited year; LSI represents (Latent Semantic Indexing) and LLR represents (Log Likelihood Ratio).

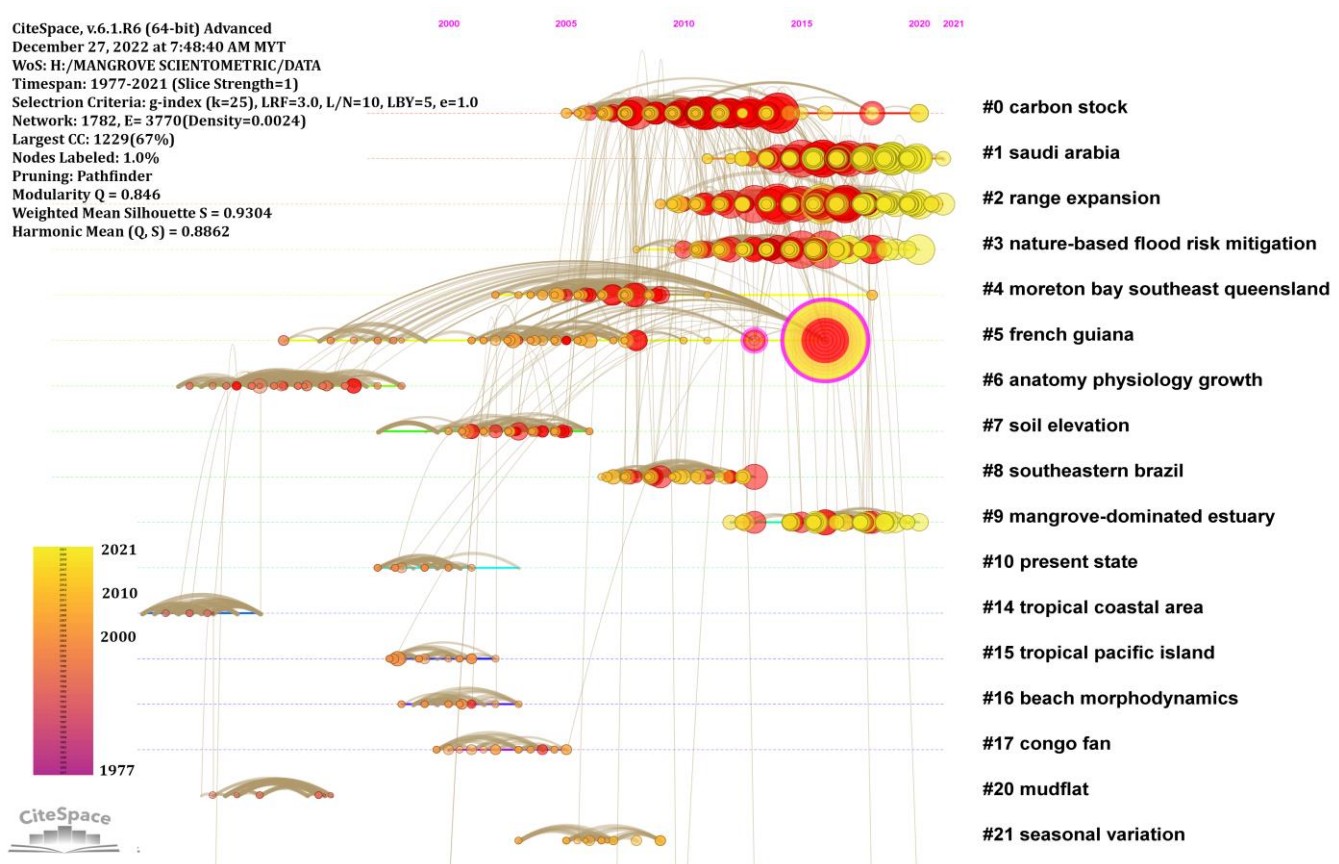

**Figure 7.** Timeline co-citation cluster analysis. Nodes represent reference names, whereas lines represent connections between those references. The size of the nodes in the network reflects the frequency of citation, while the colors of the nodes, ranging from magenta (1977) to yellow (2021), indicate the progression of research over time. References with strong citation bursts are shown with red nodes.

The fourth-largest cluster (#3) has 124 members and a silhouette value of 0.908, labelled as sea-level rise by both LLR and LSI. The major citing article of the cluster is Osland et al. [55] where they quantified the distribution of surface elevation table-marker horizon (SET-MH) stations along the Gulf of Mexico coast (USA), taking into consideration the gradients of temperature, precipitation, elevation, and relative sea-level rise. The study area included the coasts of all five U.S. states along the northern Gulf of Mexico (i.e., Florida, Alabama, Mississippi, Louisiana, and Texas). They suggested a need for long-term data of more than ten years for modelling and monitoring purposes, specifically looking at ecologically-relevant abiotic gradients at both local and regional scales.

The fifth largest cluster (#4) has 87 members and a silhouette value of 0.951, labelled as Moreton Bay southeast Queensland, by LLR, sea-level rise by LSI. The primary cited article of this cluster is Kumara et al. [56], which reported on the survival, growth, accumulation of aboveground biomass, dynamics of the sediment surface elevation, and accumulation of nitrogen in mangrove sediments in their investigation of the effects of sea level rise. Vertical accretion and surface elevation change were used to calculate the sea level rise and fall. Moreton Bay Marine Park is home to approximately 35 of the 65 species of mangroves found globally [57]. The complexity of this thriving habitat and the rich biodiversity it supports have piqued researchers' interest. Seven species of seagrass, covering 189 km$^2$, thrive in Moreton Bay [58]. By 2050, increasing sea levels, high tides, storm surges, and waves, particularly from tropical cyclones and east coast lows, are anticipated to cause accelerated coastal erosion and increased floodings of some low-lying areas in the Moreton Bay region [59].

Timelines for document co-citation analyses are useful for explaining the window of time during which a study was most widely covered by academics (Figure 7). From 2010 to 2021, there have been bursts in citations for research clusters on (#0) "carbon stock", (#1) "Saudi Arabia", (#2) "range expansion", (#3) "nature-based flood mitigation" and (#9) "mangrove-dominated survey". Taken together, these studies highlight the expanding focus on mangroves to sequester carbon and its roles in mitigating floods. Due to its resistance and tolerance to ocean warming, mangroves' expansion is closely related to climate change. Under the effects of climate change, these regions are anticipated to become local hotspots for mangrove dissemination, development, range expansion, and displacement of salt marsh [55]. Increased terrestrial C storage due to expanding mangroves has the potential to act as a significant cooling effect on a global scale [60].

### 3.5. Highly Cited Publications Based on Co-Citation Analysis

CiteSpace's visualized analysis of 4458 publications yielded a co-citation network (frequency of two different documents are cited together in other documents) with 1782 nodes, and 3770 links or connections indicate co-citations between nodes [61]. The larger the node, the higher a document is cited, demonstrating its impact on mangrove and climate change research. In the present document co-citation clustering analysis, the relationship between the top contributing references and the research clusters was mapped (Figure 8). It illustrates that the most influential references are representative of a number of key clusters, including "carbon stock", "seasonal variation", "soil elevation", and "range expansion". The evolution of mangrove and climate change research over the last three decades was analyzed using a co-citation analysis. The results were tabulated and separated into three-time frames to understand the trends better. Table 4 represents the major citing articles from 2010 to 2021, Table 5 represents articles from 2000 to 2010, and Table 6 represents articles from 1990 to 2000. This analysis provides valuable insights into the historical trend of research on this topic and can be used to predict future research directions and advancements.

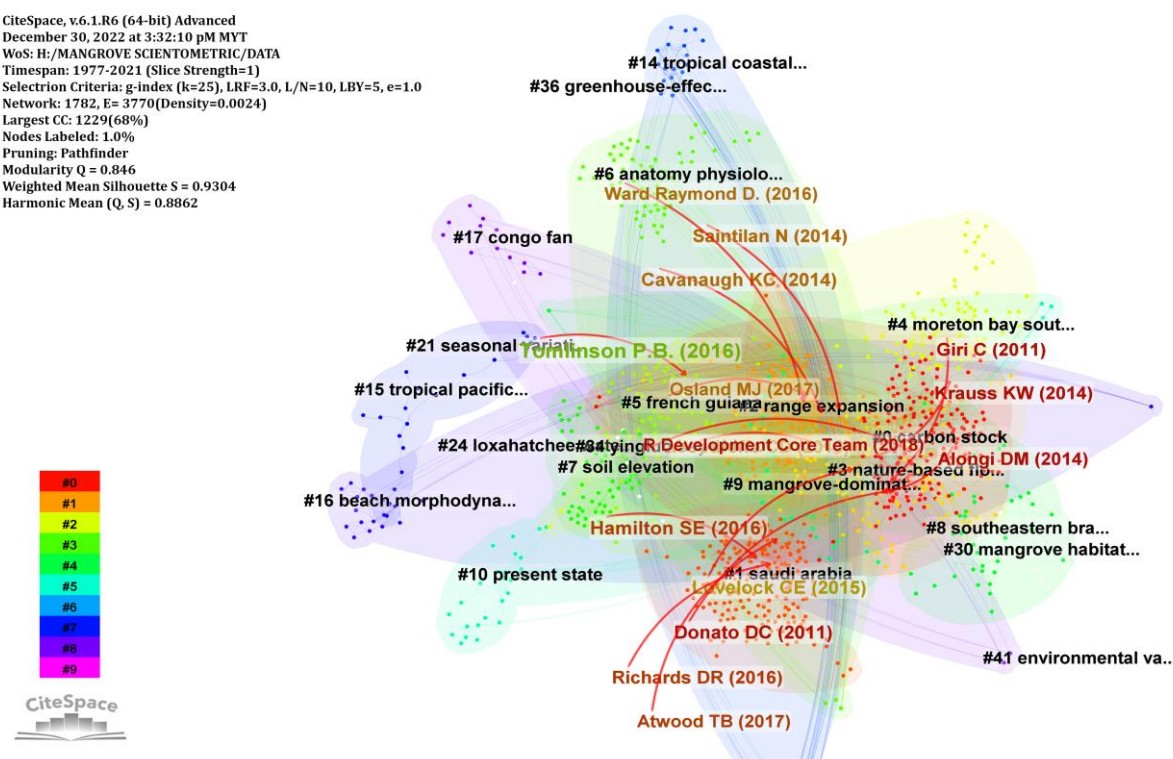

**Figure 8.** Document co-citation clustering analysis for mangrove and climate change publications.

**Table 4.** Co-citation analysis of major citing articles pertinent to mangrove and climate change (2010–2021).

| Title | References | Cluster | Citation Count | Journal |
|---|---|---|---|---|
| "The Botany of Mangroves" | Tomlinson [62] | #6 | 227 | *Cambridge University Press* |
| "Creation of a high spatio-temporal resolution global database of continuous mangrove forest cover for the 21st century (CGMFC-21)" | Hamilton and Casey [63] | #2 | 143 | *Global Ecology and Biogeography* |
| "The vulnerability of Indo-Pacific mangrove forests to sea-level rise" | Lovelock et al. [64] | #3 | 128 | *Nature* |
| "Mangroves among the most carbon-rich forests in the tropics" | Donato et al. [65] | #0 | 117 | *Nature Geoscience* |
| "Poleward expansion of mangroves is a threshold response to decreased frequency of extreme cold events" | Cavanaugh et al. [66] | #2, #3 | 115 | *PNAS* * |
| "Rates and drivers of mangrove deforestation in Southeast Asia, 2000–2012" | Richards and Friess [67] | #6 | 114 | *PNAS* * |
| "How mangrove forests adjust to rising sea level" | Krauss et al. [33] | #3 | 114 | *New Phytologist* |
| "Carbon cycling and storage in mangrove forests" | Alongi [32] | #0 | 112 | *Annual Review of Marine Science* |
| "Status and distribution of mangrove forests of the world using earth observation satellite data" | Giri et al. [49] | #2 | 111 | *Global Ecology and Biogeography* |
| "Mangrove expansion and salt marsh decline at mangrove poleward limits" | Saintilan et al. [68] | #2 | 97 | *Global Change Biology* |

\* PNAS: Proceedings of the National Academy of Sciences.

**Table 5.** Co-citation analysis of major citing articles pertinent to mangrove and climate change (2000–2010).

| Reference | Cluster | Citation Count | Journal | Article Summary |
|---|---|---|---|---|
| **Waycott et al.** [69] | #10 | 247 | *PNAS* * | In terms of biodiversity loss, seagrass meadows are right up there with mangroves, coral reefs, and tropical rainforests. |
| **Loarie et al.** [70] | #10 | 161 | *Nature* | Mangrove forests had the highest index of the velocity of temperature change (km $yr^{-1}$). |
| **Alongi** [19] | #10 | 104 | *Environmental Conservation* | A review of the global mangrove forests' condition with concluding remarks that linked the fate of mangrove forests after 2025 with technology and ecological, genetics, and forestry modelling. |
| **Chmura et al.** [71] | #0 | 92 | *Global Biogeochemical Cycles* | Data on carbon sequestration by mangroves and salt marshes from the western and eastern coasts of the Atlantic and Pacific Oceans, the Indian Ocean, the Mediterranean Sea, and the Gulf of Mexico. |
| **Erwin** [72] | #2, #6 | 90 | *Wetlands Ecology and Management* | A policy paper which discusses the significance of effective long-term restoration and management strategies for wetlands worldwide. |
| **Halpern et al.** [73] | #10 | 57 | *Conservation Biology* | The rocky reef, coral reef, hard-shelf, mangrove, and offshore epipelagic ecosystems were identified as the most at-risk in a quantitative survey among subject-matter experts. |
| **McKee et al.** [74] | #6, #7 | 48 | *Global Ecology and Biogeography* | In response to rising sea levels, the mangrove forests typical of the Caribbean have adapted by storing sediment at their roots' base. |
| **Gratwicke and Speight** [75] | #6 | 37 | *Journal of Fish Biology* | Tropical marine habitats had a higher number of species and greater habitat complexity, according to the habitat assessment score (HAS). |
| **Amiro et al.** [76] | #0 | 32 | *Journal of Geophysical Research: Biogeosciences* | Net ecosystem production (NEP) carbon loss from all ecosystems was revealed by eddy covariance measurements of carbon dioxide flux from North American forests. |
| **Cahoon et al.** [77] | #9, #6 | 27 | *Journal of ecology* | Peat collapse in mangrove forests on the islands of Guanaja and Roatan, Honduras, was brought on by changes in sediment elevation and accretion dynamics after Hurricane Mitch. |

* PNAS: Proceedings of the National Academy of Sciences.

**Table 6.** Co-citation analysis of major citing articles pertinent to mangrove and climate change (1990–2000).

| Reference | Cluster | Citation Count | Journal | Article Summary |
|---|---|---|---|---|
| **Duarte and Cebrián** [78] | #0 | 51 | *Limnology and Oceanography* | As marine ecosystems shift from being dominated by phytoplankton to angiosperms, the proportion of NPP used within the systems and consumed by herbivores decreases, while the proportion of NPP stored in sediments increases. |
| **Nicholls et al.** [79] | #7, #2 | 25 | *Global Environmental Change* | The general circulation model (GCM) scenarios for global sea level rise demonstrated that, in the absence of an adaptive response, even a relatively small global rise in sea level could have significant negative effects on mangrove forests. |
| **Stirling et al.** [80] | #6 | 21 | *Earth and Planetary Science Letters* | The timing of the end of the last interglacial period is constrained by a unique regressive reef sequence at Mangrove Bay, according to a report on the ages of eight last interglacial fossil reefs along Western Australia's continental margin. |
| **Furukawa et al.** [81] | #7 | 15 | *Estuarine, Coastal and Shelf Science* | Middle Creek mangrove swamp in Cairns, Australia was studied for its tidal currents and it was discovered that the spring flood tide trapped suspended sediment from coastal waters. The clay was selectively trapped by the mangrove's flocculation of finer particles. |
| **Hoorn** [82] | #5 | 15 | *Palaeogeography, Palaeoclimatology, Palaeoecology* | Sedimentological and palynological evidence suggests that the Guyana Shield was the primary contributor of sediment to the basins of northwestern Amazonia during the Early Miocene. |
| **Kautsky et al.** [83] | #0, #9 | 14 | *Aquaculture* | Acidification brought on by shrimp farms in mangrove environments can reduce disease resistance either directly or indirectly by causing heavy metals to be released from sediments. |
| **Smith et al.** [84] | #3 | 19 | *BioScience* | Following Hurricane Andrew, the potential interaction between two different scales of disturbance (hurricanes and lightning strikes) within mangrove forest systems was evaluated. |
| **McKee** [85] | #7 | 11 | *Journal of Ecology* | Soil redox potentials and interstitial water sulphide concentrations influenced the distributions of two dominant mangrove species in a neotropical forest. Reducing soil conditions and sulphide decreased root oxygen concentrations significantly. |
| **Ellison and Stoddart** [86] | #10 | 9 | *Journal of Coastal Research* | The stratigraphic record of mangrove ecosystems during sea-level fluctuations during the holocene indicates that low islands will be especially susceptible to the loss of mangrove ecosystems during the projected relative sea-level rise over the next 50 years. |
| **Hemminga et al.** [87] | #0 | 6 | *Marine Ecology Progress Series* | Carbon flux measurements taken in Gazi Bay, Kenya show a strong correlation between the POM fluxes of the mangrove forest and the seagrass meadows that border it. |

The top 10 highly cited publications were represented by four major research clusters as—#0—"carbon stock", #2—"range expansion", #3—"sea-level rise", and #6—"anatomy physiology growth". The highest cited publication in terms of citation count was by Tomlinson (1986, 2016), entitled "The Botany of Mangroves". The publication introduced mangroves, including their taxonomy, habitat-specific features, reproduction, and socioeconomic value. This was followed by Hamilton and Casey [63], who successfully developed a Global Database of Continuous Mangrove Forest Cover for the 21st Century (CGMFC-21) to map the global mangrove forest coverage for 2000–2012. The paper's focus highlighted

several key issues, including carbon stocks, climate change, biodiversity, food security, coastal livelihoods, and conservation-related issues that impacted the mangrove cover. Despite tremendous progress in the rest of the world, the concern is within the Southeast Asia region, where the rapid expansion of aquaculture has led to substantial mangrove deforestation [67].

Two publications on the list specifically highlighted the impact of sea-level rise on mangroves. One of them was published in Nature by Lovelock et al. [64], describing the impact of sea-level rise on Indo-pacific mangrove forests. They developed a model that calculated the survivability of mangroves in elevated sea levels by comparing the mean sea level to astronomical tide data within the region. Based on their modelling data, they were able to conclude that the survivability of Indo-pacific mangrove forests due to sea-level rise is strongly correlated to the availability of suspended matter that helps in the soil-surface elevation. It was observed that river damming significantly affected the sediment supply to mangrove forests, directly affecting the survivability of mangrove forests within this region; for example, over 80% reduction of sediment supply in the Chao Phraya River Delta has caused a significant impact on the mangrove forest in Thailand. The next publication by Krauss et al. [33] is a review published in the New Phytologist describing the adaptability strategies of mangrove forests to rising sea levels. They demonstrated the ecological attributes of mangroves that controlled the elevation gains or losses in mangrove forests. Aerial roots help in slowing the velocity of water, promoting sedimentation. Next is the elevation gain achieved by the litterfall accumulation on the soil surface, followed by the benthic mat formation from microbial and algal decomposition. The structural characteristics of mangrove roots have also been recorded as adaptation strategies in maintaining soil elevation. They also highlighted environmental and climatic impacts such as rainfall variability and elevated $CO_2$ levels to contribute to the elevation change that could impact the mangrove forest.

Two publications represented the impact of "carbon stock" on mangrove forests. The first was a publication in *Nature Geoscience* by Donato et al. [65], showing a correlation between carbon emission and mangrove loss. Their data showed that mangrove forests in the tropics are among the most carbon-dense forests dominated by the soil carbon pools found in below-ground storage. The second publication reviews the mangrove forests' ability to capture and store carbon by Alongi [32]. Seven summary points were highlighted in the review, which included (i) mangrove forests being highly productive—the carbon stock of mangrove forests was 956 tCha$^{-1}$, which was almost four times higher than rain forests (241 tCha$^{-1}$), (ii) mangroves actively capture sediment particles averaging to about 58% of the total soil carbon, (iii) mangroves contribute to an average 10%–15% of the total carbon sequestration in the ocean, (iv) mangrove deforestation will have a detrimental impact on the ecological aspects of the planet, (v) more than half of the total mangrove litterfall contribute dissolved organic carbon towards adjacent coastal zones, (vi) groundwater from the forest delivers to about 70% dissolved inorganic carbon material to adjacent waters, (vii) over 90% of the mangrove gross primary production is by canopy respiration, surface soil and mangrove waterways respiration.

Table 5 presents a co-citation analysis of major citing articles on topics related to mangroves and climate change during the period 2000–2010. The articles summarized in the table highlight various impacts of climate change on mangroves, including the loss of biodiversity, temperature change, the global condition of mangrove forests, carbon sequestration, restoration and management of wetlands, risk assessment of ecosystems, adaptation of mangrove forests to rising sea levels, habitat assessment, carbon loss from ecosystems, and the impact of natural disasters such as hurricanes on peat collapse in mangrove forests. The articles, which were published in different journals, cover a wide range of domains in the mangrove and climate change research field. These domains include biodiversity loss and risk assessment of ecosystems, temperature change and global condition of mangrove forests, carbon sequestration and ecosystem production, restoration

and management of wetlands, adaptation of mangrove forests to climate change, habitat assessment and species diversity, and the impact of natural disasters on mangrove forests.

Table 6 presents the co-citation analysis of major citing articles relevant to mangroves and climate change between 1990–2000. The articles pertain to the domains of marine ecosystems, global sea level rise, geology and reef sequences, tidal currents and suspended sediment, soil conditions and root oxygen concentrations, carbon flux measurements, and the impacts of climate change on mangrove ecosystems.

Recent, ongoing bursts can infer future patterns in an area; hence burst detection may be utilized extensively to investigate research trends [88]. "Citation bursts" demonstrate correlations between publications and sudden increases in citations. CiteSpace allows for two kinds of burst detection: citation-based and occurrence-based [89]. This study also applied this algorithm to generate the latest citation burst for examining the growing trends of climate's effect on mangrove studies. In total, 25 papers were categorized to represent various future trends. Table 7 provides a summary of the top ten future directions based on the recent surge in citations.

**Table 7.** Top 10 references with the strongest citation bursts.

| References | Year | Strength | Begin | End |
|---|---|---|---|---|
| Donato et al. [65] | 2011 | 57.45 | 2012 | 2016 |
| Giri et al. [49] | 2011 | 50.99 | 2012 | 2016 |
| Tomlinson [62] | 2016 | 37.56 | 2016 | 2019 |
| Alongi [32] | 2014 | 33.37 | 2016 | 2019 |
| Alongi [90] | 2008 | 32.88 | 2009 | 2013 |
| Mcleod et al. [91] | 2011 | 31.72 | 2012 | 2019 |
| Krauss et al. [33] | 2014 | 30.03 | 2015 | 2019 |
| Gilman et al. [92] | 2008 | 28.82 | 2009 | 2013 |
| Lovelock et al. [64] | 2015 | 28.35 | 2017 | 2021 |
| Cavanaugh et al. [66] | 2014 | 28.08 | 2015 | 2019 |

Six out of the top 10 publications with the highest citation burst in a period were also members of the highest cited articles (Table 4). They were articles by Donato et al. [65], Giri et al. [49], Tomlinson [62], Alongi [32], Krauss et al. [33], Lovelock et al. [64], and Cavanaugh et al. [66]. Six out of the ten articles presented had a citation burst that ended relatively present (2019–2021), which suggests a surge in publications and research topics pertaining to them which included Tomlinson [62], Alongi [32], Mcleod et al. [91], Krauss et al. [33], Lovelock et al. [64] and Cavanaugh et al. [66]. These six articles with recent burst covered three main research clusters, which included #0—"carbon stock", #3—"sea-level rise", and #6—"anatomy physiology growth". The highly cited article by Mcleod et al. [91] reviewed the role of coastal vegetation in sequestering $CO_2$. They described the role of coastal ecosystems, including the mangroves, salt marshes and seagrasses, in global carbon sequestration. It is widely accepted that more than half of the C-sequestered in the mangrove, seagrass beds, and salt marshes originated externally.

### 3.6. Keywords Co-Occurrence Analysis

Co-occurrence analysis is based on the principle of counting the frequency with which a set of keywords appears in the same document, clustering these words based on the number of occurrences, reflecting the affinity of these words, and then analyzing the structural shifts of the disciplines and topics represented by these words [93]. Keywords, primarily words and phrases that describe the fundamental ideas of articles, can be used to track the development of research areas and domains [94,95]. We used CiteSpace to create a map of keyword occurrences and relative frequency, with the strongest burst

keywords and co-occurrences highlighted and the mapping language deciphered in this study. From CiteSpace, 856 keywords and 4412 links were retrieved between 1976 and 2021. The broad array of research on mangroves was shown by many linked lines (more than the number of nodes—856) and complex linkages between keywords. Analysis of high-frequency keywords revealed 11 categories as research hotspots and frontiers in the field of climatic impact on mangroves (Figure 9): #0 "Palynology", #1 "Remote sensing", #2 "Coral reef", #3 "Blue carbon", #4 "Photosynthesis", #5 "Heavy metals", #6 "Climate change", #7 "Behaviour", #8 "Embryonic development", #9 "Coastal management", #10 "Southeast Asia". Table 8 displays the results of a clustering classification of mangrove research and development based on the meanings provided by its keywords.

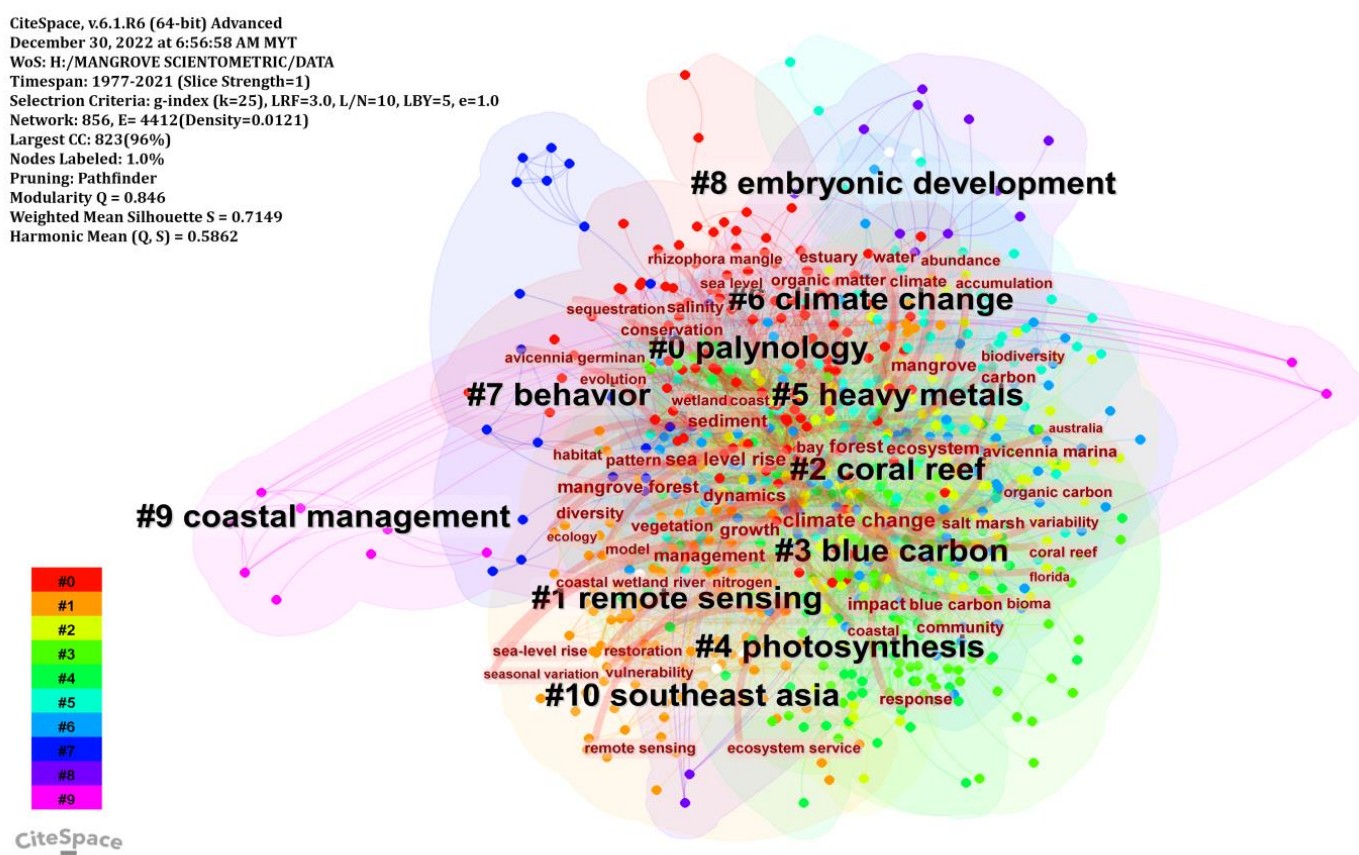

**Figure 9.** Keyword clustering analysis for mangrove and climate change publications (1977–2021).

The top 10 frequently given keywords that appeared in the 4458 articles have appeared more than 150 times (Table 9). These keywords could represent the research trends and popular topics. "Climate change", "forest", "sea level rise", "dynamics" and "ecosystem" were the five top keywords with a strength of frequency of above 370. The most recent spike was found in articles that included the keyword "climate change", with a peak occurring between the years 2018 and 2022. "Forest", "sea-level rise", "dynamics", "mangrove", "sediment", and "growth" are among the early keywords to have been heavily used.

The relative importance of the keywords is represented by the size of the node. More significance is assigned to larger nodes [96]. Nodes that are close to each other have a higher relationship [96,97]. The rapid growth and shifting boundaries of a field can be traced back to the popularity of certain keywords. Rapidly increasing keyword searches can reveal shifts at the field's cutting edge and foretell future growth patterns. These searches were shown in keyword co-occurrence networks from 1990–2000 (Figure 10), 2000–2010 (Figure 11), and 2010–2021 (Figure 12). Figure 13 presents the historical summaries and evolution paths of emerging research topics in mangrove–climate change

studies based on the keyword analysis. The keywords appearing on the timeline have been listed chronologically and based on their frequency of usage. This timeline provides an overview of the development of research topics related to the impact of climate change on mangroves over the past thirty years. Notably, during 1990 to 2000, the research focus was on keywords such as "growth", "evolution", "dynamics", "sediment", and "hydrodynamics". From 2000 to 2010, research topics shifted towards keywords such as "nitrogen", "carbon", "seasonal variation", "organic matter", and "climate change". Finally, from 2010 to 2022, research interest evolved towards keywords such as "blue carbon", "sequestration", "expansion", "carbon stock", "wave attenuation", "accumulation rates", and "coastal vulnerability". These trends in keyword usage over time reflect the changes in research priorities and provide valuable insights into the ongoing scientific discourse and knowledge advancement in the field of mangrove–climate change studies.

**Table 8.** Most frequent keyword label describing the cluster label for mangrove and climate change literature (1977–2021).

| Cluster | Cluster Label | Keyword Label |
|---|---|---|
| #0 | Palynology | Sediment, organic matter, estuary, climate, sea level, evolution, model, coastal, coast, environmental change, record, marine sediment, system, gulf, sea, Holocene, basin, indicator, continental shelf |
| #1 | Remote sensing | Vulnerability, sea-level rise, adaptation, classification, accretion, protection, area, establishment, coastal erosion, delta, regeneration, landscape, gradient, recovery, rehabilitation, protected areas, time series |
| #2 | Coral reef | Dynamics, ecosystem, impact, mangrove forest, management, pattern, conservation, community, diversity, response, restoration, ecosystem service, biodiversity, abundance, ecology, assemblage, resilience, population, island |
| #3 | Blue carbon | Organic carbon, wetland, biomass, variability, sequestration, soil, storage, productivity, carbon stock, land use, freshwater, stock, emission, deforestation, coastal ecosystem, carbon dioxide, decomposition, exchange, nutrient, biogeochemistry, flux |
| #4 | Photosynthesis | Forest, growth, salinity, *Avicennia marina*, nitrogen, *Rhizophora mangle*, temperature, stable isotope, plant, gas exchange, drought, nutrient enrichment, tolerance, *Laguncularia racemosa*, seedling, stress, mangrove plant |
| #5 | Heavy metals | Bay, carbon, water, accumulation, seasonal variation, water quality, transport, surface sediment, trace metal, environmental impact, contamination, mangrove sediment, pollution, phosphorus, geochemistry |
| #6 | Climate change | Sea level rise, mangrove, vegetation, salt marsh, coastal wetland, Florida, Australia, expansion, Gulf of Mexico, dispersal, black mangrove, wave attenuation, seagrass, carbon storage, tidal marsh, eutrophication, deposition, mangrove expansion |
| #7 | Behaviour | Ocean, density, pacific, inundation, bird, science, precipitation, movement, level rise, intertidal, |
| #8 | Embryonic development | Crab, fiddle crab, hypoxia, dissolved oxygen, canonical correspondence analysis, fatty acid, habitat quality, ocypodidae, biochemical composition, decapod, consumption |
| #9 | Coastal management | Chlorophyll, fluorescence, cotton, energy change, dissipation, transformation, mangrove forest, ATP, adenylate kinase, adenylate energy change |

**Table 9.** Top 10 frequently used keywords for mangrove and climate change publications (1977–2021).

| Keywords | Start Year | Frequency | Burst Begin | Burst End |
|---|---|---|---|---|
| climate change | 2000 | 871 | 2002 | 2018 |
| forest | 1992 | 660 | 1996 | 2010 |
| sea level rise | 1992 | 427 | 1996 | 2007 |
| dynamics | 1993 | 403 | 1996 | 2007 |
| ecosystem | 1996 | 375 | 1999 | 2010 |
| mangrove | 1993 | 356 | 1995 | 2008 |
| impact | 2003 | 331 | 2002 | 2007 |
| sediment | 1997 | 312 | 1996 | 2010 |
| mangrove forest | 2001 | 310 | 2005 | 2010 |
| growth | 1992 | 282 | 1996 | 2007 |

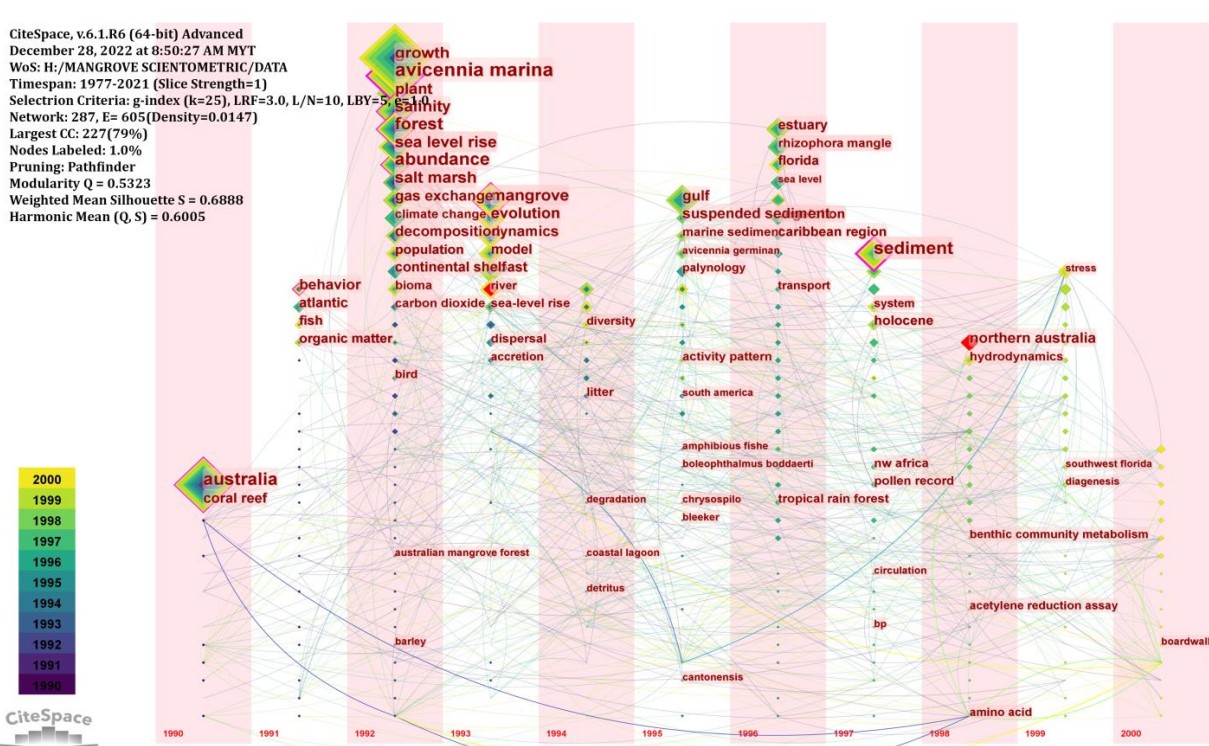

**Figure 10.** Keyword co-occurrence network from 1990 to 2000 for mangrove and climate change-related publications.

### 3.7. Dual Map Overlay

To gain a comprehensive understanding of the evolution of the field, we performed a dual map overlay analysis using CiteSpace. This analysis provides an interactive representation of the relationships between disciplines and the distribution of individual publications from different organizations [98]. The dual map overlay of journals illustrates the subject distribution of journals, with the left side of the graph displaying citing journals and the right side indicating cited journals. The citation relationships between articles in cited journals and articles in cited journals are represented by colored lines [99,100]. A change in the trajectory of citations from one region to another can be used to identify the influence of articles from a different discipline. The depth of these trajectories, as measured by the Z-score, reflects the accumulated citations of the works [98]. Figure 14 provides insight into the citation patterns of works on the impact of climate change on mangrove

forests. The dominant citing region, Ecology, Earth, Marine, was found to have influenced two distinct domains, Earth, Ecology, Geophysics (Z-score of 4.17) and Plant, Ecology, Zoology (Z-score of 7.74).

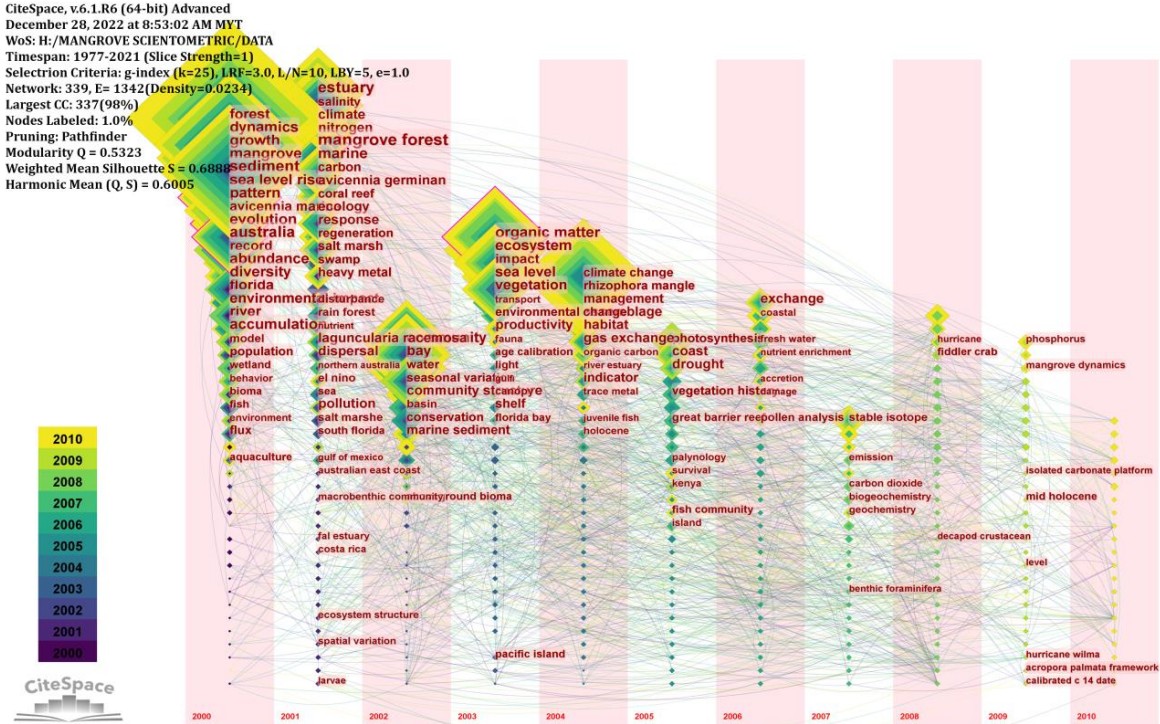

**Figure 11.** Keyword co-occurrence network from 2000 to 2010 for mangrove and climate change-related publications.

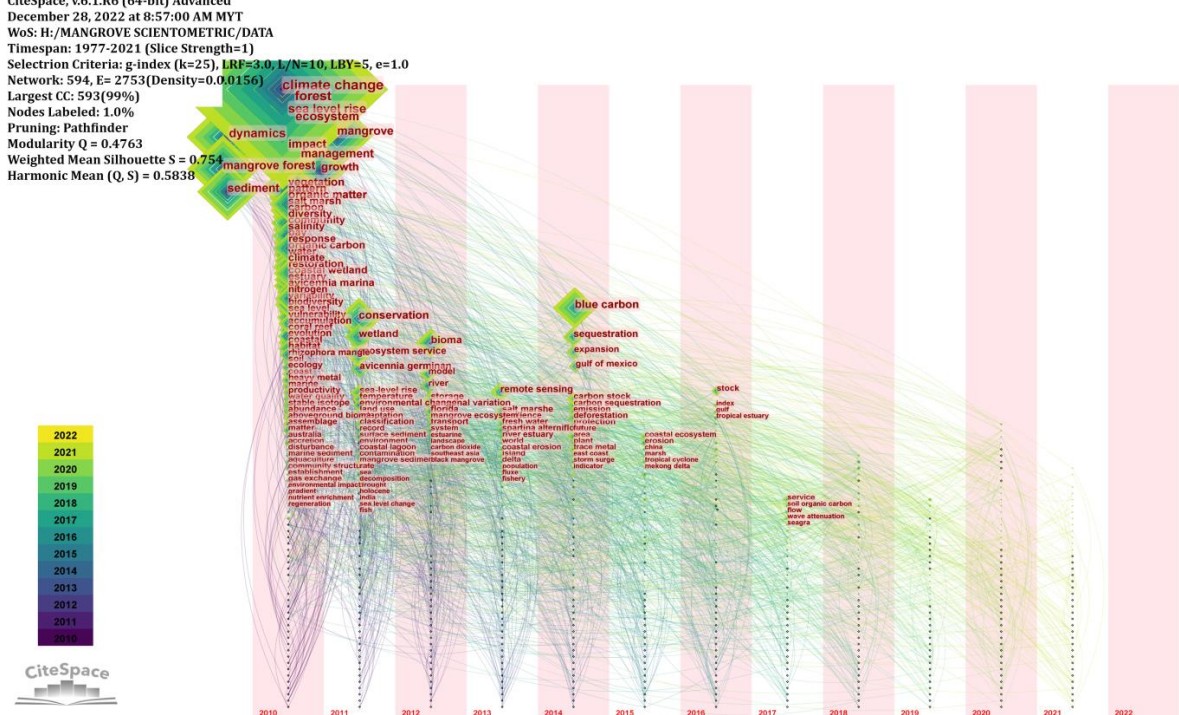

**Figure 12.** Keyword co-occurrence network from 2010 to 2021 for mangrove and climate change-related publications.

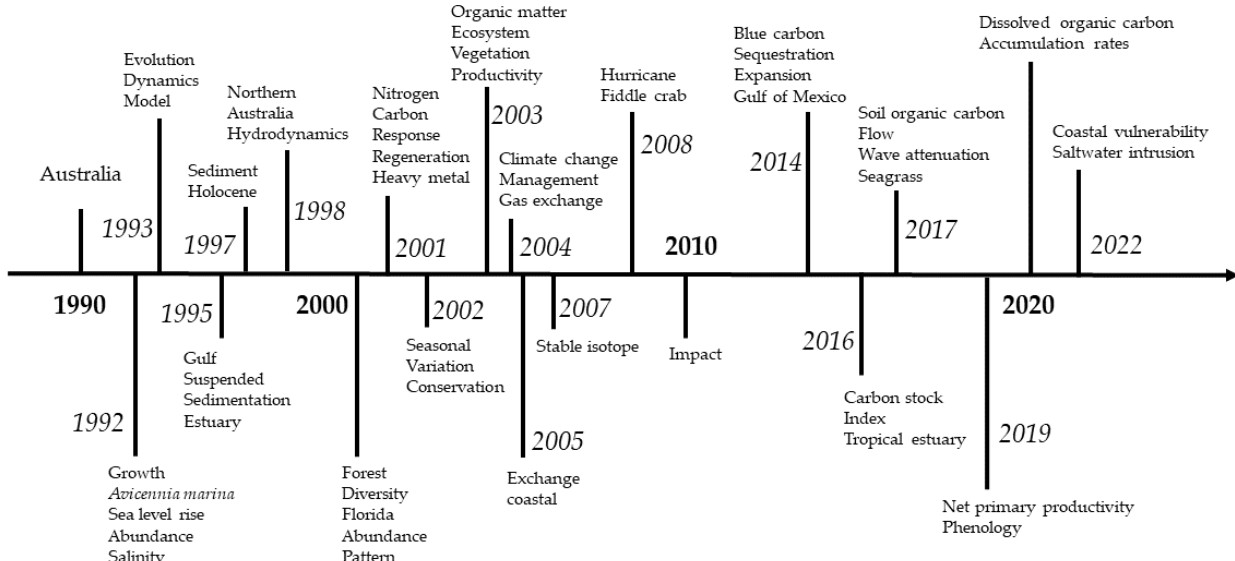

**Figure 13.** Evolution of research topics in mangrove-climate change studies from 1977 to 2021.

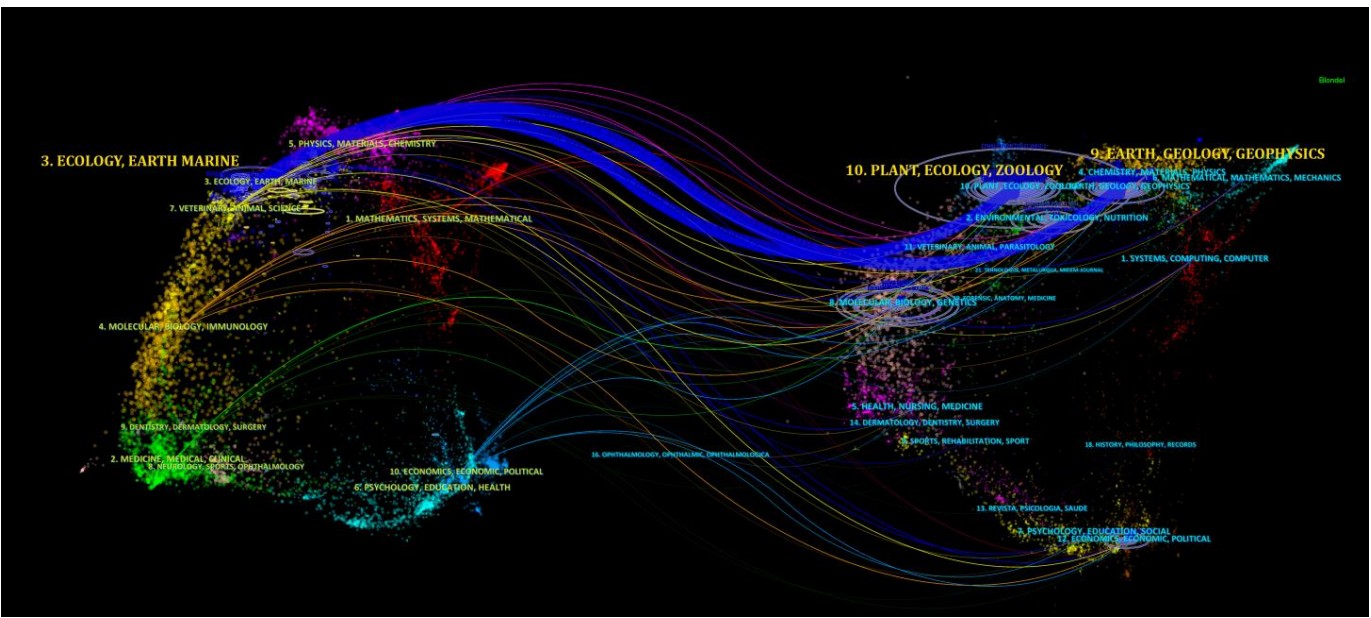

**Figure 14.** Domain-level citation patterns in mangrove and climate change-related research from 1977 to 2021.

## 4. General Discussion

The global average yearly rate of mangrove loss is estimated to be 1–2%, with losses during the last quarter century ranging from 35–80% [92]. This is likely due to relative climatic risks such as sea-level rise as well as non-climate-related anthropogenic stresses [20,21,101–103]. We must address several crucial questions to enhance our comprehension of the fate of mangrove forests on the cusp of global climate change: (1) the impact of climate change on mangroves; (2) mangrove resilience and adaptation strategies; and (3) policies and regulations for mangrove conservation.

### 4.1. The Impact of Climate Change on Mangroves

By serving as coastal greenbelts [104], the mangroves are able to cope-up naturally with a few consequences of climate change from their adaptive nature [64]. Hence, these wetlands were recognized as part of the Ecosystem-based Adaptation (EBA) for climate

change in the United Nations Framework Convention on Climate Change (UNFCCC) nearly a decade ago [105,106]. However, several perturbations associated with climate change (e.g., extreme heat, high precipitation, frequent storms, and sea level rise) remain a threat to the mangrove cover worldwide [107,108]. Mangroves may be significantly altered by climate change; one potential impact on mangroves is an expansion of their range, as rising sea levels and warming temperatures may create more suitable habitats for these trees to thrive. The global poleward expansion of mangrove range limits exemplifies the expansion and contraction of species' geographic ranges resulting from climate change [68,109]. Reduced frequency and intensity of extreme freeze events brought on by climate change are predicted to alter biotic interactions, leading to the tropicalization of some formerly temperate ecosystems as tropical biota expand poleward and displace temperate biota [110,111]. Similar processes are likely to govern the distribution and abundance of freeze-sensitive organisms along the subtropical coasts (e.g., China, Australia, New Zealand, western Mexico, southern Africa, and western South America).

An increasing number of studies have confirmed that mangroves are expanding their range as a direct result of global warming. For instance, mangrove forests, specifically the black mangrove (*Avicennia germinans*), are expanding in the Gulf of Mexico states of Texas, Louisiana, and Florida [112]. Cavanaugh et al. [66] found a correlation between the significant expansion of mangroves in Florida between 1984 and 2011 and a decline in the frequency of discrete cold occurrences. They detected a decline in mangrove cover that only occurred as the temperature dropped below the −4 °C threshold for mangrove forests. Similarly, a study in the Pacific Islands found that mangroves are also moving further inland as sea levels rise and that the rate of this inland migration is increasing [113]. Other studies have also documented the expansion of mangrove ranges in response to climate change in various locations around the world, including in Latin America [68], Africa [114], and Asia [115]. In many instances, these range expansions have been accompanied by alterations in the composition and diversity of mangrove ecosystems, as some species may be more or less able to adapt to the altered conditions [116,117].

One of the most important impacts of climate change on mangrove forests is sea level rise. Mangrove forests are at risk of flooding and extinction as a result of rising sea levels [33,64]. The summary of the detrimental effects of sea-level rise due to climate change is shown in Table 10.

**Table 10.** Effects of sea-level rise on mangrove forests.

| Effect | Description | Reference |
|---|---|---|
| Habitat loss | Due to the trees' inability to survive in newly flooded areas or for extended periods, rising sea levels can result in the disappearance of mangrove forests. The numerous species, including fish, crabs, and birds that depend on mangroves for habitat, may suffer as a result. | Loucks et al. [118]; Li et al. [119] |
| Modifications in ecosystem function | Mangrove forests are essential to the well-being and operation of coastal ecosystems. The ecosystem as a whole may be negatively impacted by the loss of mangroves brought on by sea level rise. | Lovelock and Ellison [59] |
| Coastal erosion | By providing natural protection from storms and waves, mangrove forests serve as a buffer against coastal erosion. Increased erosion, damage to the coastal margins, and harm to dependent communities may result from the loss of mangroves brought on by sea level rise. | Thampanya et al. [26] |
| Displacement of communities | Rising sea levels may force communities living near mangrove forests to relocate, leading to social and economic disruption. | Gilman et al. [120]; Barua et al. [121] |

Coastal management and residing communities must create alternative strategies and consider them to deal with the risks posed by sea level rise to lessen these effects. This might entail taking steps like erecting sea walls, creating flood-protection barriers, or moving communities to higher ground [122]. However, the construction of physical barriers may also negatively impact the existing mangrove cover [123]. Coastal wetland ecosystems are highly sensitive to changes in precipitation regimes, as indicated by numerous studies. For instance, the relationship between rainfall variables and mangrove expansion was found to be positive in Moreton Bay, Australia, with changes in rainfall being the key factor driving the rate of landward encroachment [124]. On the northern coast of the Persian Gulf and Oman Sea, reduced precipitation of 43%, increased evaporation and salinity from the year 2000 resulted in low mangrove cover [92,125–127]. Research conducted in the Iranian province of Qeshm found that variations in precipitation significantly affected the development of mangroves there [128]. Similarly, climatic models and decadal climatic trends of Brazil's extensive mangrove forests are predicted to experience drought stress as a result of rising temperatures and falling precipitation [129,130]. In contrast, a sudden drop in average annual rainfall did not impact the Net Ecosystem Production (NEP) in the Pichavaram mangrove ecosystem in India [131], but higher rainfall levels were found to impact the NEP and caused a decrease in soil effluxes in western Everglades National Park, Florida [132]. Similarly, Liu and Lai [133] estimated that heavy precipitation reduced the Gross Primary Productivity (GPP) level in a humid Hong Kong mangroves by 32.6%. The effects of changes in precipitation patterns on mangrove forests brought on by climate change are shown in Table 11.

**Table 11.** Precipitation variability pattern on mangrove forests.

| Effect | Description | Reference |
|---|---|---|
| Drought | Irrespective of their ability to grow and thrive, drought can stress mangrove trees. In some instances, a protracted drought can even cause mangrove trees to perish. | Mafi-Gholami et al. [125,127] |
| Flooding | Mangrove forests may also suffer from increased flooding brought on by changes in precipitation patterns. Coastal flooding can result in environmental degradation and ecological imbalance. Consequences include mangrove forest migration inland or seaward, sedimentation and biodiversity threat. | Munji et al. [134]; Wong et al. [135] |
| Alterations in nutrient availability | Variations in precipitation patterns can also impact the availability of nutrients in mangrove soils. For instance, increased precipitation may wash nutrients away, while drought may make water-soluble nutrients less available. | Hilaluddin et al. [136] |

Examining the complex and multifaceted effects of carbon dioxide ($CO_2$) in the atmosphere on mangrove forests is another crucial impact of climate change. The relationship between elevated atmospheric $CO_2$ levels and mangrove growth has been explored and found to be species-specific. The location of the mangroves and the interactive effects of $CO_2$ with salinity and nutrient availability can play a significant role in the mangrove's response. Changes in species patterns within estuaries may occur as a result of different species' abilities to respond to the changing drivers [137]. Experiments indicate that mangrove growth is enhanced in the presence of additional nutrients, and conversely, a decline in growth may result in decreased utilization and a potential buildup of dissolved soil nutrients [138].

The potential of Indonesian mangrove forests to serve as a tool for global climate change mitigation was assessed by Murdiyarso et al. [103]. The study estimated the carbon stocks of these forests to be approximately over 1000 MgCha$^{-1}$, translating to an average of 3.14 PgC when scaled to the country's mangrove extent of 2.9 Mha. Despite this significant carbon storage capacity, Indonesia has lost 40% of its mangroves over the past three decades, primarily due to aquaculture development, resulting in annual

emissions of 0.07–0.21 Pg $CO_{2e}$. Although mangrove deforestation only accounts for 6% of Indonesia's total forest loss [139], reversing this trend would result in a reduction in emissions equivalent to 10–31% of the estimated annual emissions from land-use sectors. Similarly, the potential of Sundarban mangroves as a tool for mitigating global climate change was investigated by Rodda et al. [140]. These tidal forests, known to be the world's largest block of halophytic mangrove forests, were found to be a net carbon sink with a mean net primary production of 276 $gCm^{-2}\,yr^{-1}$. The study analyzed the carbon balance and seasonal dynamics of this pristine mangrove ecosystem and found that the variations in $CO_2$ fluxes are largely influenced by environmental factors such as temperature, vapour pressure deficit, rainfall, and active photosynthetic radiation. The study conducted by Van Vinh et al. [141] analyzed the seasonal variability of $CO_2$ emissions in Can Gio, the largest mangrove forest in Vietnam. The findings indicated that the initial rainfall pulse of the monsoon season resulted in the highest $CO_2$ emissions from both the soil and tree trunks, likely due to an increase in ecosystem photosynthesis and a decrease in ecosystem respiration. The study also suggested that the persistent high temperatures prevalent in Southern Vietnam contributed to these elevated $CO_2$ emissions. A study on $CO_2$ and $CH_4$ emissions within a *Rhizophora* spp. mangrove forest in New Caledonia found that $CO_2$ emissions were highly variable and primarily influenced by tides [142]. $CO_2$ emissions were higher during spring tides compared to neap tides, which was attributed to increased microbial activity within the soil and greater exchange surface between the soil and water column. The effects of increased atmospheric $CO_2$ on mangrove forests are shown in Table 12.

**Table 12.** Increased atmospheric $CO_2$ variability on mangrove forests.

| Effect | Description | Reference |
|---|---|---|
| Increased photosynthesis | Through the process of photosynthesis, mangrove trees are able to absorb and store atmospheric $CO_2$. Increased photosynthesis and higher rates of carbon sequestration in mangrove forests may result from higher atmospheric $CO_2$. | Reef et al. [143] |
| Alterations in nutrient cycling | Increasing atmospheric $CO_2$ may change the nutrients available in mangrove soils, which may impact the development and survival of mangrove trees. | Lovelock et al. [59]; Alongi [137] |
| Changes in water availability | Increasing atmospheric $CO_2$ concentrations may cause water cycle changes affecting mangrove forests' water availability. This may affect the development and survival of mangrove trees. | Lovelock et al. [59]; Driever et al. [144] |
| Alterations in atmospheric and oceanic temperatures | Increasing atmospheric $CO_2$ may be a factor in rising global temperatures, which may have an effect on mangrove forests by altering sea level and the frequency and severity of extreme weather events. | Ellison [145]; Lovelock and Ellison [146] |

*4.2. Adaptation Strategies and Resilience of Mangroves in Connection to Climate Change*

Mangroves are capable of tolerating a diverse range of environmental conditions, including variable water levels, exposure to saltwater, and fluctuations in temperature. Research has established that mangroves can dissipate up to 76% of wave energy and reduce wind velocity by 50% [147,148], making them effective in providing protection against hurricanes and storms. However, mangroves are also susceptible to the impacts of climate change, such as rising sea levels, increased frequency and intensity of storms, alterations to water quality, and long-term changes to their structure and composition [92,149–152]. Despite these challenges, mangroves exhibit physiological traits that enhance their resilience to extreme weather events, such as large nutrient reserves, rapid nutrient turnover rates, and tolerance to inundation and salinity. These adaptations allow mangrove species to recover from storm damage through the re-sprouting of epicormic shoots [148,153].

The resilience of mangrove ecosystems to adapt to changing sea levels by migrating inland has been well documented. However, the expansion of coastal developments and the corresponding increase in human activity impede this natural migration, making it crucial to monitor changes in surface elevation within mangrove ecosystems [33,152]. The exposure of low-lying coastal communities to the effects of sea level rise, driven by factors such as demographic and settlement trends and anthropogenic subsidence, has become growing concern [21,154]. Human activities, such as deforestation, conversion of upstream areas to agriculture, urban development, and aquaculture, can significantly increase sedimentation within mangrove ecosystems, which can negatively affect their productivity. Increased sedimentation can weaken mangrove trees, bury their aerial roots, and create a favourable environment for the invasion of terrestrial plants [155,156]. Furthermore, the inflow of freshwater can alter mangrove ecosystems' salinity and result in species composition changes [157]. Development activities, such as dams, reservoirs, and sand mining, can also reduce the sediment supply to mangroves and cause subsidence, further threatening their survival [158]. As sea levels continue to rise, mangroves are retreating inland in response. However, as human activity continues to expand further inland, the available area for mangroves to retreat is shrinking, presenting an even greater threat to their continued existence [156,159]. The mangroves' adaptation strategies against climate change's impacts are shown in Table 13.

**Table 13.** Adaptation strategies and resilience of mangrove forests.

| Effect | Description | Reference |
|---|---|---|
| Vertical accretion | One way that mangroves can adapt to rising sea levels is through a process called vertical accretion, in which sediment is deposited on the surface of the forest floor. This can help to keep pace with rising sea levels and prevent the mangroves from being inundated. | MacKenzie et al. [160] |
| Seed dispersal | Mangroves reproduce via nautohydrochory, a process that allows seeds to float on water and be transported to new locations. As sea levels rise or their habitat changes, this enables them to colonize new landward areas. | Wijayasinghe et al. [161] |
| Genetic diversity | Mangroves with a high level of genetic diversity have a better chance of adapting to shifting environmental conditions. These ecosystems' resilience can be improved by conserving a variety of mangrove species and populations. | Duke et al.[162]; Arnaud-Haond et al. [163] |
| Restoration and conservation efforts | Planting new mangrove seedlings and restoring damaged mangrove forests can help increase these ecosystems' resilience. Protecting existing mangroves from development and other threats can also help to ensure their long-term survival. However, the approaches for (re)establishing mangroves have advantages and disadvantages that must be carefully considered before implementation. | Jones et al. [164]; Zimmer et al. [165] |

### 4.3. Policies and Regulations for Mangrove Conservation

Multiple initiatives and policy frameworks have been established to aid in the conservation of mangrove forests. The Convention on Wetlands (Ramsar Convention), signed in 1971, provides a framework for the preservation and responsible use of wetlands, including mangroves. By designating wetlands of international importance and adopting measures for their conservation and management, the parties of the convention aim to preserve these important ecosystems [166]. Examples of Ramsar sites that have had a positive impact on mangrove conservation include the Xuan Thuy Wetland Reserve in Vietnam [167], Sungai Pulai Forest Reserve in Malaysia [168], Pichavaram mangrove in India [169], Koh Kapik in Cambodia [170], and the Gulf of Montijo Ramsar Site in Panama [171].

The United Nations Framework Convention on Climate Change (UNFCCC), signed in 1992, aims to stabilize greenhouse gas concentrations in the atmosphere. Mangroves play a role in mitigating the effects of climate change by serving as a sink for carbon dioxide. As a result, the UNFCCC includes provisions for conserving and enhancing carbon sinks, including mangroves, through initiatives such as REDD+ [172].

The International Union for Conservation of Nature (IUCN) has developed Mangrove Management Guidelines, providing recommendations for the sustainable use and protection of these ecosystems [173]. The Global Mangrove Alliance, a partnership of over 30 organizations, seeks to conserve and restore mangroves around the world. With the support of over 100 specialists in mangrove research, finance, and policy, the alliance aims to increase the number of protected mangroves, improve the management of existing forests, and promote sustainable use. The alliance's goal is to reduce carbon dioxide emissions and restore 20% of the world's mangroves by 2030 [174,175].

In addition to these international initiatives, several national and regional policies and frameworks are in place to support mangrove conservation. These can include laws and regulations protecting mangroves, conservation and restoration funding, and programs promoting the sustainable use of mangroves. To improve mangrove forest governance, Indonesia, home to the world's largest mangrove regions, has adopted several creative laws and initiatives that have resulted in a large-scale mangrove conservation and restoration programme [176,177]. Nationally Determined Contributions (NDC) is an example of its policy that has resulted in the government expanding the role of forestry in climate change mitigation. The notion of blue carbon is being adopted, and mangrove forests have been added to Indonesia's national greenhouse gas inventory and are now being evaluated as an alternative natural climate solution [177].

## 5. Conclusions

Along with standardization of the methods for mangrove inventory, and carbon assessment, the research in line to predict and face the challenges of climate change is crucial. According to Rogers et al. [178], some of the actions that can be taken to ensure that mangroves continue to flourish include benchmarking and improving mangrove health and extent, reducing inconsistent mangrove governance, and prioritizing the landscape for restoration. To ensure the success of scientific projects and policies, findings from the research should be disseminated among interested parties and encouraged by citizen scientists [179,180]. The research findings are to be applied without compromise to ongoing management initiatives for any necessary modifications.

**Author Contributions:** Conceptualization, T.C.S. and M.N.A.; methodology, T.C.S. and F.L.; software, T.C.S. and N.H.A.R.; validation, B.S., I.G., V.R. and J.B.; formal analysis, Z.V.-G. and M.N.A.; investigation, Z.V.-G. and J.B.; resources, I.G., F.L. and B.S.; data curation, J.B.; writing—original draft preparation, T.C.S. and F.L.; writing—review and editing, M.N.A.; visualization, T.C.S.; supervision, B.S.; project administration, M.N.A.; funding acquisition, M.N.A., I.G., V.R., Z.V.-G. and J.B. All authors have read and agreed to the published version of the manuscript.

**Funding:** This research was funded by the Department of Higher Education, Ministry of Higher Education Malaysia under the LRGS program (LRGS/1/2020/UMT/01/1; LRGS UMT Vot No. 56040) entitled 'Ocean climate change: potential risk, impact and adaptation towards marine and coastal ecosystem services in Malaysia'. The work was also supported by the PASIFIC program GeoRecoproject funding from the European Union's Horizon2020 research and innovation programme under the Marie Sklodowska-Curiegrant agreement No.847639 and from the Ministry of Education and Science.

**Institutional Review Board Statement:** Not related.

**Informed Consent Statement:** Not applicable.

**Data Availability Statement:** No data was generated from the study.

**Conflicts of Interest:** The authors declare no conflict of interest.

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
