# Peer review of "Mapping the Link between Climate Change and Mangrove Forest: A Global Overview of the Literature"

_forests, doi:10.3390/f14020421_

Round 1

Reviewer 1 Report

I found that the article is interesting, highlighting mangrove-climate change issues using scientometric analysis. I only have few comments:

- Results: I'd suggest to have a historical summary of the trends of emerging topics in mangrove-climate change. It can be derived from keywords (Figure 9-12), which is then split into timeline.

- Discussion: mangroves can adapt with SLR by shifting landwards and increasing soil elevation. This capacity is influenced by human activities, for example human settlements on the coastline that might hinder mangroves to grow landward when sea-level is rising, or soil elevation deficit due sediment retention by dams. Further discussion on this issue in section 4.2 Adaptation Strategies will be worthwhile as it has correlation with policies or regulation at national/sub-national level (section 4.3), for example coastal management.

line 26: Do you mean: Mangrove has a vital role.... ?

line 62-63: Please briefly specify what are the 'review articles' ?

line 84, 598: I prefer to omit words 'etc, and many more...'

line 474: I think SLR is also included.

line 484: is it for all forest types or only wetland ?

Author Response

Dear Reviewer,

We would like to thank you for taking the time to review our manuscript titled "[Mapping the link between climate change and mangrove forest: A global overview of the literature]."

We have carefully considered all of your comments and have made the necessary revisions to the manuscript. Please find attached a detailed response to your comments and suggestions.

Thank you once again for your time and effort in reviewing our manuscript.

Results: I'd suggest to have a historical summary of the trends of emerging topics in mangrove-climate change. It can be derived from keywords (Figure 9-12), which is then split into timeline.

Response 1: We have addressed these concerns in our study by including a detailed historical summary of emerging topics in mangrove-climate change in Figure 13. The trends and evolution of these topics are presented through a timeline based on the analysis of keywords in Figures 9-12.

-Discussion: mangroves can adapt with SLR by shifting landwards and increasing soil elevation. This capacity is influenced by human activities, for example human settlements on the coastline that might hinder mangroves to grow landward when sea-level is rising, or soil elevation deficit due sediment retention by dams. Further discussion on this issue in section 4.2 Adaptation Strategies will be worthwhile as it has correlation with policies or regulation at national/sub-national level (section 4.3), for example coastal management.

Response 2: We have addressed the issue you raised by including a more detailed discussion of mangrove adaptation strategies and resilience in connection to climate change in Section 4.2. (line 630 – 644).

line 26: Do you mean: Mangrove has a vital role.... ?

Response 3: Statements have been ammended to : Mangroves play a crucial role in maintaining the stability of coastal regions, particularly in the face of climate change. (line 26)

line 62-63: Please briefly specify what are the 'review articles' ?

Response 3: Satements ammended : There have also been several highly cited review articles on mangroves, such as the review on mangrove ecosystems and rehabilitation [14], mangrove carbon dynamics [30], so-cio-economics, ethnobiology and management of mangroves [31], carbon cycling and storage [32] and impact of rising sea levels on mangroves [33]. (line 56-59)

line 84, 598: I prefer to omit words 'etc, and many more...'

Response 4: All of those terms have been removed

line 484: is it for all forest types or only wetland ?

Response 5: We have rephrase the sentence to bring more clarity

Reviewer 2 Report

General Comment:

This manuscript used a scientometric software, CiteSpace, to analyze the trend and connections of literature published through the year 1977-2021, regarding the climate change on mangrove wetlands. It is interesting to learn about the research trend during the last four decades and I appreciate the authors’ work. This is an important and informative analysis that indicates the future research direction in the context of “how climate change impacts mangroves”.

Given the manuscript’s present structure and context, I politely suggest a major revision for further review.

Specific comments:

1)      Overall, the major concerns I have are mostly related the visualization. 

General issues for all figures:

(1) The texts in the upper left corner are too small and hard to read. If it is needed, please increase the font.

(2) The magnitude or axis texts on color bars are not clear. Please modify.

(3) Please revise all figure/table legends 

(4) Last but not least, the resolution needs improvement; especially the Figure 13, I can’t read anything.

2)      Figure 1, how do you get the n = 4558; I assumed there are 100 papers that fall into at least two “excluded records” groups. Please clarify. (6311-126-1029-698 is not 4558)

3)      Figures 4 & 5 color bars are not clear and I don’t understand what those colors mean. 

4)      Section 3.3, These five research disciplines (Line 198) seem too wide. Could you provide some specific examples?

5)      Lin 227, which figure/table shows the “MI”.

6)      Figure 6,8,9 The color of the color bar and map don’t match.

7)      Table 3, typo in the notation below table “Si1”, “Si2”

8)      In Figure 7, axis text (e.g., year) is too small to read (see details below). The color bar is not clear. And what is the meaning of the last sentence Legend? What does the magenta ring mean in the figure?

9)      Figure 8 color bar and its texts are too small. 

10)  Check the unit thoroughly. Please use the correct format/super-/sub-script, e.g., t C ha-1, C (temperature), CO2, etc.

11)  Section 3.5, it is good to have these results based on the data from 1977-2021. Importantly, I would like to see the results based on data within different decades as the author did in section 3.6 (line 426-432). Lots of recent critical publications may be hidden due to short years. I think these new publications are more representative indicating the latest research trend and future research direction.

12)  In the Discussion section, it is great the authors conclude and discuss the results based on three sub-sections. However, I would like to read more deepened discussions instead of “listing” the information. I suggest the authors summarize the content for each sub-sections and make the discussions concise. All that information formatted in “bullets” can go into tables as evidence/examples to support the points discussed here. 

Author Response

Dear Reviewer,

We would like to thank you for taking the time to review our manuscript titled "[ Mapping the link between climate change and mangrove forest: A global overview of the literature]."

We have carefully considered all of your comments and have made the necessary revisions to the manuscript. Please find attached a detailed response to your comments and suggestions.

Thank you once again for your time and effort in reviewing our manuscript.

1)      Overall, the major concerns I have are mostly related the visualization.

General issues for all figures:

  • The texts in the upper left corner are too small and hard to read. If it is needed, please increase the font.
  • The magnitude or axis texts on color bars are not clear. Please modify.
  • Please revise all figure/table legends
  • Last but not least, the resolution needs improvement; especially the Figure 13, I can’t read anything.

Response 1: We have made the necessary amendments to increase clarity and readability, including increasing font sizes, modifying axis texts, revising figure legends, and improving image resolution. The magnitude and axis texts on color bars have also been modified to make them more clear and easy to read. Amendments have been made to Figure (4 – 12, 14)

2)      Figure 1, how do you get the n = 4558; I assumed there are 100 papers that fall into at least two “excluded records” groups. Please clarify. (6311-126-1029-698 is not 4558)

Response 2: We apologize for any confusion caused by the incorrect value of 4,558 presented in Figure 1. The correct value is 4,458, and this has been rectified in the updated version of the figure and all throughout the manuscript.

3)      Figures 4 & 5 color bars are not clear and I don’t understand what those colors mean.

Response 3: We have made the necessary amendments to improve the image resolution and increasing the font sizes.

4)      Section 3.3, These five research disciplines (Line 198) seem too wide. Could you provide some specific examples?

Response 4: We have addressed the issue you raised by including a more detailed descriptions (Line 193 -207)

5)      Line 227, which figure/table shows the “MI”.

Response 5: We have made ammendments to the statement (Line 225-230)

6)      Figure 6,8,9 The color of the color bar and map don’t match.

Response 6: We have made necessary adjustments to the image to address these issues.

7)      Table 3, typo in the notation below table “Si1”, “Si2”

Response 7: We apologize for the error

8)      In Figure 7, axis text (e.g., year) is too small to read (see details below). The color bar is not clear. And what is the meaning of the last sentence Legend? What does the magenta ring mean in the figure?

9)      Figure 8 color bar and its texts are too small.

Response 8&9: We have made necessary adjustments to the image to address these issues.

10)  Check the unit thoroughly. Please use the correct format/super-/sub-script, e.g., t C ha-1, C (temperature), CO2, etc.

Response 7: We have rectified all the errors that were identified in our manuscript.

11)  Section 3.5, it is good to have these results based on the data from 1977-2021. Importantly, I would like to see the results based on data within different decades as the author did in section 3.6 (line 426-432). Lots of recent critical publications may be hidden due to short years. I think these new publications are more representative indicating the latest research trend and future research direction.

Response 11: We have now included a similar analysis in Section 3.5, as you suggested. The analysis is separated into three different-time frames (1990-2000, 2000-2010, and 2010-2021) to better understand the trends over the last three decades. We have also provided the citing articles from each time frame in Tables 4, 5, and 6, respectively.

12)  In the Discussion section, it is great the authors conclude and discuss the results based on three sub-sections. However, I would like to read more deepened discussions instead of “listing” the information. I suggest the authors summarize the content for each sub-sections and make the discussions concise. All that information formatted in “bullets” can go into tables as evidence/examples to support the points discussed here.

Response 12: We have rewritten the discussions with more detailed and comprehensive analyses, and we have also provided a summary of the content for each sub-section to make the discussions more concise and easier to follow.

In addition, we have tabulated the summaries accordingly, as you suggested, to provide evidence and examples to support the points discussed in the manuscript.

Round 2

Reviewer 2 Report

Some minor typos/suggestions:

1)    Figure 7: remove years on the top of the figure if it is not needed

2)    You may want to keep Tables 4,5, and 6 in the same format/organization

3)    Figures 11, 12: years on the color bar are wrong

4)    Figure 13: What are the topics for 2020 and 2021? if you intend to show them.

Author Response

Thank you for your feedbacks, please refer to the attachment for the detailed response.

  1. Figure 7: remove years on the top of the figure if it is not needed

Response 1: Thank you for your feedback on Figure 7. Unfortunately, the image is generated by the CiteSpace software and does not come with an option to edit out the years at the top of the figure. The years represent the progression of each cluster and could be beneficial for some readers as it provides a timeline for the clusters.

  1. You may want to keep Tables 4,5, and 6 in the same format/organization

Response 2: We have made the necessary edits to ensure that all the tables (4,5 and 6) are now in the same format.

  1. Figures 11, 12: years on the color bar are wrong

Response 3: We apologize for the error. We have made the necessary ammendements to Figure 11 and 12.

  1. Figure 13: What are the topics for 2020 and 2021? if you intend to show them.

Response 4: We have made the necessary ammendements to Figure 13.